# Improving Large Language Model Planning with Action Sequence Similarity

**Xinran Zhao**[1,2*], **Hanie Sedghi**[1], **Bernd Bohnet**[1], **Dale Schuurmans**[1], **Azade Nova**[1]
[1]Google DeepMind,  [2]Carnegie Mellon University

## Abstract

Planning is essential for artificial intelligence systems to look ahead and proactively determine a course of actions to reach objectives in the virtual and real world. Recent work on large language models (LLMs) sheds light on their planning capability in various tasks. However, it remains unclear what signals in the context influence the model performance. In this work, we explore how to improve the model planning capability through in-context learning (ICL), specifically, what signals can help select the exemplars. Through extensive experiments, we observe that commonly used problem similarity may result in false positives with drastically different plans, which can mislead the model. In response, we propose to sample and filter exemplars leveraging plan side action sequence similarity (AS). We propose **GRASE-DC**: a two-stage pipeline that first re-samples high AS exemplars and then curates the selected exemplars with dynamic clustering on AS to achieve a balance of relevance and diversity. Our experimental result confirms that GRASE-DC achieves significant performance improvement on various planning tasks (up to ~11-40 point absolute accuracy improvement with 27.3% fewer exemplars needed on average). With **GRASE-DC**[*]+**VAL**, where we iteratively apply GRASE-DC with a validator, we are able to even boost the performance by 18.9% more. Extensive analysis validates the consistent performance improvement of GRASE-DC with various backbone LLMs and on both classical planning and natural language planning benchmarks. GRASE-DC can further boost the planning accuracy by ~24 absolute points on harder problems using simpler problems as exemplars over a random baseline. This demonstrates its ability to generalize to out-of-distribution problems.

## 1 Introduction

Planning is important for intelligent agents when exploring the environment and conducting complex multi-hop actions to achieve their goals strategically. Classical studies in planning mainly leverage search-based algorithms and reinforcement learning to tackle these problems. Recent advances in utilizing Large Language Models (LLMs) as the backbone of agents, e.g., for games (ToT, Yao et al., 2023) and travel scheduling (Xie et al., 2024), call for the need to improve model planning ability to facilitate various downstream applications.

Recent work achieves good performance on LLM planning with a combination of search-based algorithms and LLM decoding (Besta et al., 2024; Silver et al., 2024; Lehnert et al., 2024); however, multiple rounds of prompting in a tree structure, e.g., Monte-Carlo Tree Search (MCTS), can lead to high inference cost (Yao et al., 2023). To further improve the effectiveness and efficiency, this paper focuses on improving the planning capability of LLMs with direct prompting in the in-context learning (ICL) (Brown et al., 2020) manner. We aim to seek signals that help select the good demonstrative task-plan examples in the context, i.e. exemplars (Rubin et al., 2022). Previous work (Ye et al., 2023) in the natural language processing (NLP) community considers an exemplar selector as a dense retriever (Karpukhin et al., 2020) that matches the semantics of two task descriptions. However, for planning tasks, semantically similar task descriptions with one different initial state can lead to different core strategies and eventually drastically different correct plans. For example,

---

*  Work done as a student researcher at Google DeepMind. Correspondence to xinranz3@andrew.cmu.edu.

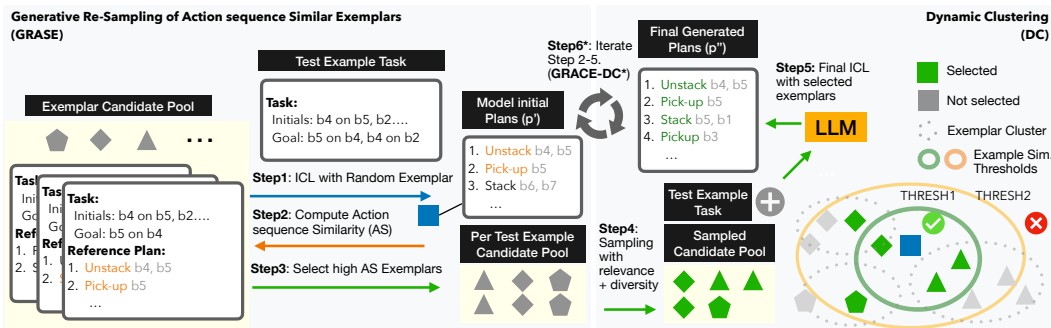

Figure 1: An illustration of our two-stage GRASE-DC pipeline. Given a test example, [GRASE stage:] we first use the random exemplars from the candidate pool to acquire the initial model plan. We then utilize actions in this plan to rank the pool with the action sequence similarity. [DC stage:] we further sample the specific exemplar pool for the test example with the relevance and diversity in the lens of clusters based on action sequence similarity. Finally, we conduct ICL and prompt LLMs with the sampled pool and original test example. We can iteratively apply GRASE-DC, i.e., GRASE-DC*, by re-sampling exemplars with the action sequences of the generated plans.

requiring an end state of a block on the top or bottom of a 100-block pile will only have a one-word difference in task description: *put b1 on the table vs. put b100 on the table*, but the problem complexity, as well as the desired operations in plans, is different. To avoid potential false positive exemplars, we explore the signals from the actual plans with the view of their essence: *a sequence of ordered and dependent actions*. We validate such intuition by comparing the ICL performance on various tasks with exemplars sampled from signals in the tasks or plans. We propose to consider plans as a series of ordered actions and measure the **A**ction sequence **S**imilarity (AS). Specifically, we measure the similarity between two action sequences with their longest common action sequence (LCAS) normalized by sequence lengths. Our analytical experiments with Oracle plans of test examples confirm that the proposed AS is a better and more robust signal in selecting exemplars to assist the model planning, compared to random sampling or signals from task descriptions.

We propose **GRASE-DC** a two-stage pipeline that empirically leverages AS to improve LLM planning with ICL. As shown in Figure 1, we first acquire the model's initial output plans for the test examples with randomly selected exemplars. Then, we perform the first stage, **G**enerative **Re**-sampling of **A**ction **S**equence **S**imilar **E**xemplars (**GRASE**): we utilize the model-generated plans to compute the action sequence similarity with all exemplar candidates. Last, given the set of similar exemplars sampled in GRASE, we further remove potential noise and redundancy with a **D**ynamic **C**lustering (**DC**) step: we capture the geometric relations among exemplar candidates plus each test example with clustering over their distance as the reciprocal of AS. We utilize these relations to keep a new dynamic set of exemplars for each individual test example with a balance between *relevance* and *diversity*. We use this final set of exemplars, with a dynamic size for each test example, to construct the prompt for standard ICL to guide models to generate plans. Since GRASE-DC maintains the original ICL pipeline, with the final model-generated plans, we can iteratively apply the GRASE and DC steps, we denote this iterative approach as GRASE-DC*.

We evaluate the performance of **GRASE-DC** on both classic planning benchmarks (Aeronautiques et al., 1998; Höller et al., 2020; Bohnet et al., 2024) as well as natural language planning benchmarks (Zheng et al., 2024). Through extensive experiments, we show that GRASE-DC achieves significant and robust performance improvement in planning through ICL with LLMs across tasks with a concise set of exemplars. GRASE-DC achieves up to ~11-40 point absolute accuracy improvement and 27.3% fewer exemplars needed on average, compared to random exemplar selection. GRASE-DC* with validator further achieves similar or better performance with fewer exemplars, compared to GRASE-DC, with up to 18.9% performance improvement and 39% fewer exemplars.

We further analyze the cost of selecting exemplars with our approach against random selection. We show that with parallel computation of plan similarity, the extra compute cost for selecting exemplars with AS is negligible compared to random, and GRASE only requires only one additional prompt to the LLMs, which is much less than the overhead of the search-based algorithms such as MCTS and ToT. We further investigate how to use MLP and BPE-Proxy to approximate AS at different performance-efficiency trade-offs. We estimate the floating-point operations per second

(FLOPs) needed for exemplar selection and plan generation for each test example. The proposed MLP achieves 95% performance of GRASE with around 66% FLOPs; BPE-Proxy achieves 83% performance of GRASE with around 27% FLOPs, showing the effectiveness of these methods.

To assess the broader applicability of our method, we further evaluate it on different backbone LLMs as well as data beyond its original training distribution. We observe that GRASE consistently enhances the planning capabilities of different backbone LLMs. This means it reliably leads to better performance across a range of LLMs, regardless of their design. Furthermore, GRASE-DC significantly outperform random baseline (~24 absolute points) on harder and out-of-distribution test examples using simple exemplar candidates.

## 2 METHODOLOGY

### 2.1 TASK FORMATION

In this paper, we investigate how to improve the planning capability of large language models (LLMs) through in-context learning (ICL). Given the task description of the planning problem (e.g., the initial component states and the final goal), we prompt the model with a set of exemplar candidates and let the model directly output the executable and verifiable plans in the response.

Formally, each planning instance, either an exemplar candidate ($c$) or test example ($t$), contains a task description and a referential Oracle plan, i.e., $(c_d, c_p)$ refers to an exemplar candidate and $(t_d, t_p)$ refers to a test example. Each plan $c_p$ or $t_p$ can be rewritten into a sequence of actions $\mathcal{A} = \{a_1, a_2, ...\}$. For each task, there exists a validator, e.g., a rule-based system VAL (Howey et al., 2004), to verify if each action in $\mathcal{A}$ is valid considering its impact on the states and if the objectives (described as the object states) in task descriptions are satisfied after executing a plan[1]. It is common to exist multiple plans that satisfy one specific task description, which can be different in the action orders or number of steps.

In ICL setting, for each test example that contains task description $t_d$ and plan $t_p$, we sample a set of exemplars $\mathcal{D}_t = \{c_1, c_2...\}$ from all exemplar candidates $\mathcal{D}$ to form the prompt for model $\mathcal{M}$ (typically a LLM) and collect the model generated plans: $p' \sim P(t_d, \mathcal{D}_t, \mathcal{M})$, where VAL can verify if $p'$ is executable and can satisfy $t_d$, where $p'$ is not necessarily the same as the referential $t_p$. For a planning task, the actions in sequence are temporally dependent, where the pair-wise order matters. To compute the similarity of two action sequences $\mathcal{A}_i, \mathcal{A}_j$, we propose to find the longest common action sequence $\mathcal{A}_{lc}$ (ordered and consecutive) that exists in both, denoted as $\text{LCAS}(\mathcal{A}_i, \mathcal{A}_j)$. Action sequence similarity of a pair of plans, as a measure of plan similarity, is then defined as $\text{Sim}_{AS} = |\text{LCAS}(\mathcal{A}_i, \mathcal{A}_j)|^2 / (|\mathcal{A}_i| \cdot |\mathcal{A}_j|)$. Next, we discuss the details of our method.

### 2.2 GRASE-DC: ICL WITH ACTION SEQUENCE SIMILARITY

We propose GRASE-DC as a two-stage pipeline that leverages $\text{Sim}_{AS}$ to help sample exemplars for conducting ICL, as shown in Figure 1.

**(1) GRASE**: **G**enerative **R**e-sampling of **A**ction Sequence **S**imilar **E**xemplars: Given a test example, we first randomly select a set of exemplars from the candidates to obtain an initial model-generated plan ($p'$). Next, in the first GRASE stage, we rewrite $p'$ into the corresponding action sequence $\mathcal{A}'$ and rank the exemplar candidates with $\text{Sim}_{AS}$ between $\mathcal{A}'$ and each $\mathcal{A}_c$. After this stage, we can already conduct conventional ICL with the ranking by selecting the Top-N (e.g., N=40) exemplars to include in the context for prompting the LLMs. However, $\text{Sim}_{AS}$ can further provide us with relational information among candidates, which can be leveraged to further refine the exemplar candidates for each test example.

**(2) DC**: **D**ynamic **C**lustering: To reduce potential redundancy and noise in the candidates we conduct dynamic clustering, with the reciprocal of $\text{Sim}_{AS}$ defining the pair-wise distance. We keep the

---

[1]Planning problems are commonly written in Planning Domain Definition Language (PDDL, Aeronautiques et al., 1998), which provides a standard representation that guarantees problem and solution verification with task domain description. For tasks not written in PDDL, there is a commonly used rule-based verification system, e.g., check all the constraints of a travel planning problem. The verification of a plan is generally considered less costly than searching for a plan.

exemplars with the highest interval of similarity (e.g., > mean plus three standard deviations) with the test example. For other examples, we cap the maximum number of exemplars per cluster to ensure a diverse set of candidates and include it in the context. With DC, we automatically collect a different set of exemplars for each test example, which relieves the need to search for the number of exemplars, i.e., N in Top-N.

GRASE-DC maintains the original pipeline of ICL, where operations are all done on the exemplar selection, not the instruction or the output, which makes the pipeline flexible and easy to iterate over. We can either iterate the GRASE step before conducting DC, or iterate GRASE and DC steps together. We denote these iterative versions GRASE* and GRASE-DC* respectively.

Next, we discuss the GRASE and DC steps in more detail. For clarity, common abbreviated terms along with explanations are gathered in Table 5 in Appendix A.12.

### 2.2.1 GRASE: GENERATIVE RE-SAMPLING OF ACTION SEQUENCE SIMILAR EXEMPLARS

Exemplar selection (Rubin et al., 2022; Zhang et al., 2022; Ye et al., 2023) has been shown to boost ICL performance by leveraging problem-level similarity (e.g., similarity in task description). In this section, we aim to build an exemplar scoring method for LLM planning. Specifically, given a set of candidate exemplars $\mathcal{D}$ and a test example description $t_d$, the method scores and ranks each entry in $\mathcal{D}$ to decide the corresponding $\mathcal{D}_t$ that then forms the prompt.

Beyond problem-level similarity, we identify that the underlying plan similarity reveals the actual similarity or the potential mutual referencing ability of two planning tasks since (1) objects in planning tasks can be renamed; (2) complex tasks can be composed of multiple threads of simple tasks, e.g., reverse the order of a three-block pile. Two similar-sized problems can share similar task descriptions but require completely different threads. We present a qualitative example comparing what exemplars will be selected via high task or plan similarity in Appendix A.10.

As a result, we treat $\text{Sim}_{AS}$ as the way to score the exemplars. In an analytical case, we propose to use the Oracle plans for the test example ($t_p$) to compute the score to validate if it acts as a good signal for exemplars. We denote this analytical method as $\text{Baseline}_{AS}$. Besides acting as a proof of concept, modeling planning performance with $\text{Baseline}_{AS}$ can also be used to detect the quality of a certain exemplar pool, which helps to decide if extra candidates are needed.

To empirically leverage $\text{Sim}_{AS}$, instead of Oracle plan $t_p$, we let the model generate $p'$ (i.e., $\mathcal{A}'$) by itself to compute the $\text{Sim}_{AS}$. The method to generate $p'$ can be arbitrary. In our experiments, we first use randomly sampled exemplars to let the model generate plans in an ICL manner. Besides starting with random sampling, GRASE can cooperate with any strategies that output model-generated plans, e.g., ICL with problem-level similarity or Chain-of-Thought prompting (Wei et al., 2022). In this direction, we extend our experiment with the model-generated plans from both GRASE (Appendix A.5) alone and GRACE-DC (Section 3.2), i.e., an iterative approach of our strategy.

As described in (Kambhampati et al., 2024), we can obtain hard critiques (if the current plan is executable or correct) from VAL outputs for PDDL planning problems to assist LLMs. Before re-sampling, we can also apply VAL and only conduct GRASE on the test examples where models generate the wrong plans initially, as a classical rejection sampling (RS) pipeline. We denote this variant as GRASE+VAL and GRASE-DC+VAL for our pipeline at different steps. We show the performance gain from RS with either GRASE and random sampling in A.8. Note that the action sequence is not an equivalent representation of the original plans since the objects are ignored. We explore other design choices for plan representation in Section A.7.

### 2.2.2 DC: DYNAMIC CLUSTERING WITH ACTION SEQUENCE SIMILARITY

One critical component that contributes to the success of ICL is the choice of exemplar candidates. In the previous section, we discuss the ranking of the candidates. In this section, we further discuss the curation of the candidate pool for noise removal.

In a realistic scenario, the exemplar candidates are either from previous interaction records of the environment, e.g., in Web Arena (Zhou et al., 2023), or automatic environment exploration, e.g., in BAGEL (Murty et al., 2024). However, since the test examples are unknown when collecting the exemplar candidates, there could be noise introduced to the context when simply selecting the Top-N

in the exemplar ranks. Specifically, we identify two kinds of noise: (1) duplicating or unnecessary examples; and (2) unnecessary or redundant actions in a good example, with potential complementary actions from other examples. The existence of these kinds of noise suggests the importance of keeping a set of exemplars considering both the *relevance* and *diversity*, with similar findings in Auto-CoT (Zhang et al., 2023). We denote the whole strategy as Dynamic Clustering (DC):

**Relevance**: The goal of this step is to remove less relevant exemplars for each test example, since for different test examples, there is no guarantee of a fixed number of good exemplars in the pool. We leverage $\text{Sim}_{AS}$ between generated plans and each exemplar candidate plan to sample a dynamic number of candidates for each test example. That is, we compute the mean plus one standard deviation for all $\text{Sim}_{AS}$ scores among the candidates and discard any candidate that scores below this threshold. We then use the remaining candidates to build the clusters.

**Diversity**: Upon acquiring the first rough candidate set with good relevance, the second step aims at sampling a diverse set of exemplars with removed duplication. We leverage the $\text{Sim}_{AS}$ between each pair of these candidates to conduct Agglomerative Hierarchical Clustering (Müllner, 2011) to capture the internal relations among candidates. Before clustering, to ensure that highly relevant exemplars are not discarded, similar to the relevance step, we keep all candidates with scores above mean plus X standard deviation. In practice, we select X to be 3.

We control the number of clusters with a hyperparameter $N_c$ [2]. With the clusters, we can sample a diverse set of exemplars to perform ICL by ensuring that there are fewer than three examples selected per cluster. $N_c$ is considered to control balance between relevance (large $N_c$) and diversity (small $N_c$). In Section 3.2, we show robust performance with different values of $N_c$, which helps alleviate the cost of deciding the hyperparameter.

# 3 EXPERIMENTS AND ANALYSIS

## 3.1 EXPERIMENTAL SETUP

**Dataset and LLM Backbone.** Similar to existing work (Valmeekam et al., 2023a; Bohnet et al., 2024), we conduct our main experiments on data collected by or created from the pipeline of (Aeronautiques et al., 1998; Höller et al., 2020). Specifically, we conduct experiments on four PDDL tasks: *BlocksWorld*, *Minigrid*, *Logistics*, and *Tetris*, with details in Appendix A.1. For natural language planning, we conduct experiments on Trip Planning (Zheng et al., 2024). We use 300 test examples for each task and the originally provided training set as our exemplar candidates. For each test example, there is an Oracle test plan given, which is a valid plan and satisfies the goal in the task description, but it is not necessarily the only viable plan. If the use of backbone LLM is not specified, we use Gemini 1.5 Pro (Gemini Team et al., 2024) as the default to generate plans at test time. We also experiment with other commercial and open-source LLMs, including GPT-4-Turbo (Achiam et al., 2023), Claude-3.0-Opus (Anthropic, 2024), and LLama 3.1 (Dubey et al., 2024) with different parameter sizes (results are in Section 3.3).

**Baseline and Metric.** For baselines, we use *Random* to denote the commonly used random sampling (Valmeekam et al., 2023a; Bohnet et al., 2024). To validate our intuition on the effectiveness of AS in Section 2.2.1, we also use task description similarity as a baseline. Given the task description of a test example ($t_d$) and an exemplar candidate ($c_d$), we compute the similarity as the token overlap with QA-F1, following (Khashabi et al., 2020; Zhao et al., 2023). We denote this method as *Task*. For the metric, similar to (Valmeekam et al., 2023a; Bohnet et al., 2024), we use planning accuracy, which denotes the portion of test examples with generated plans that are both executable at each step (no undefined behavior or failed pre-condition) and satisfy the final goals.

## 3.2 PERFORMANCE ON PDDL TASKS

**What signal helps ICL for planning?** We first validate our intuition in Section 2.2.1 by comparing the ICL performance between sampling exemplars with signals from the task description

---

[2] Empirically, with $n$ exemplar candidates to be clustered, we set the number of clusters to be $(n^{0.25}+1)*N_c$, when there are more clusters considered, typically more exemplars are included in the context for ICL.

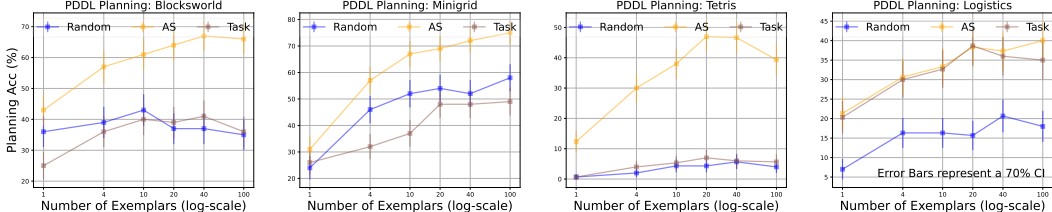

Figure 2: PDDL Planning on various tasks with Gemini 1.5 Pro. Baseline$_{AS}$ denotes ranking exemplars with the plan similarity given Oracle test plans. *Task* denotes the baseline that calculates the similarity between each test example and exemplar candidate with a token overlap in descriptions.

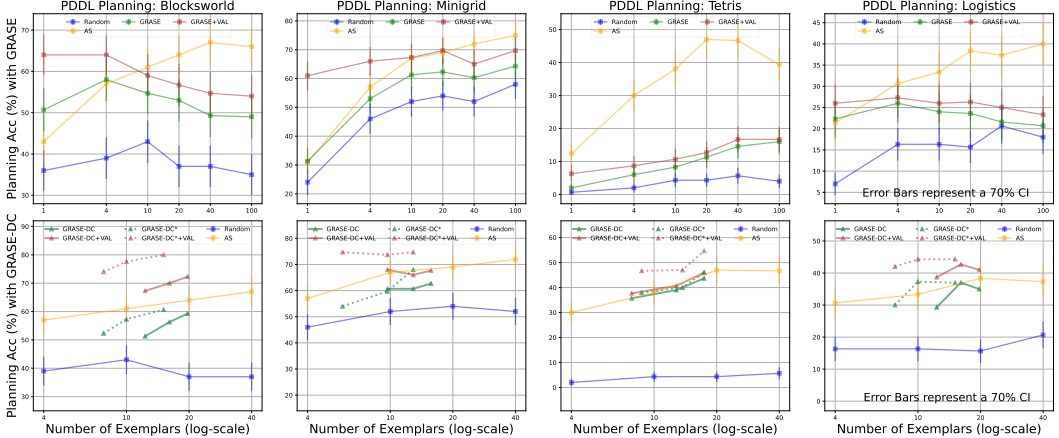

Figure 3: PDDL planning accuracy on various tasks with Gemini 1.5 Pro. Baseline$_{AS}$ (AS lines) denotes the use of Oracle test plans. Baseline$_{AS}$ and *Random* lines are the anchors across rows (the second row focuses on exemplars from 4 to 40). GRASE denotes the use of model output plans from random exemplars. VAL denotes the use of the plan validator. GRASE-DC$^*$ denotes another iteration with the model outputs from GRASE-DC. Numbers of exemplars for GRASE-DC and its iteration denote the average number of exemplars used over the whole test set with $N_c = 1, 2, 3$.

(*Task*) and plans (Baseline$_{AS}$), where both methods are based on token matching, with no contextualized embedding involved. The comparison is shown in Figure 2, we can observe that the proposed Baseline$_{AS}$ presents a strong and robust signal on selecting the exemplars, which achieves significant improvement in ICL performance with Gemini in all domains, compared to *Random* and *Task*. It is observed that the performance of *Task* over different datasets is not consistent. Specifically, while in Blocksworld and Minigrid, signals from the task descriptions (*Task*) are misleading and lead to an accuracy even worse than *Random*, it performs almost similarly to *Random* in Tetris. On the other hand, in Logistics, *Task* achieves similar performance as Baseline$_{AS}$. The reason can be that the action types are much attached to the objects, e.g., between an airplane and airports, there is only one kind of action defined: flying airplane. Examples with similar object sets, which can be captured by task descriptions, will lead to similar action sequences.

**Main results on PDDL planning tasks.** After validating the performance of Baseline$_{AS}$, we then evaluate the performance of our proposed GRASE-DC on PDDL tasks. Figure 3 shows the trend of performance change with line charts over varied numbers of exemplars. For the detailed scores, we also present the performance in tables in Appendix A.11. We also explore other empirical methods built upon Baseline$_{AS}$ in Section 3.6.

As shown in Figure 3 (upper row), with GRASE, we observe significant and consistent performance improvement against Random. GRASE achieves 15 absolute planning accuracy points improvement (43 to 58) on Blocksworld, as well as other tasks: 6.3 on Minigrid (58 to 64.3); 10.3 on Tetris (5.7 to 16); 5.3 on Logistics (20.7 to 26). If the plan validator is provided (GRASE+VAL) and the re-sampling is only done on failed examples, we can observe an extra ~3-5 accuracy points gain.

Since the generated plans themselves can be wrong, GRASE can suffer from diminishing performance gain with additional exemplars. However, to our surprise, when the number of exemplars

Figure 4: PDDL Planning on Blocksworld with various LLMs with different numbers of exemplars. All models are using the same set of exemplars for *Random*. *Opus* denotes Claude-3.0-Opus.

is small (e.g., 1 and 4 for Blocksworld), GRASE can perform better than Baseline$_{AS}$. One potential reason is that Gemini can generate plans toward a correct preferred direction even with random exemplars. However, minor mistakes can make the whole plan invalid [3]. GRASE helps Gemini retrieve and learn from the exemplars from preferred directions, thus the minor mistakes are addressed. Baseline$_{AS}$, on the other hand, helps sample exemplars with the Oracle test plans, which can deviate from the preferred directions, e.g., in action priority. We also observe consistent performance improvement over AS through iterating only over GRASE in Appendix A.5.

Next, in Figure 3 (lower row), we observe that GRASE-DC, with $N_c = 1, 2, 3$, achieves improved performance compared to the baselines. In the first iteration (solid line), GRASE-DC achieves ~11-40 point absolute planning accuracy improvement over *Random*. Similarly, VAL brings extra performance gain, with consistently improved performance over the analytical Baseline$_{AS}$. After one iteration (dashed line), GRASE-DC$^*$, either with or without VAL, helps achieve higher performance with few exemplars (pushing the curve to the upper left), which further improves the overall effectiveness and efficiency of the pipeline at inference. We also show the original performance without zoom-in (i.e., not focusing on exemplars from 4 to 40) in Figure 8 in the appendix.

## 3.3 PERFORMANCE WITH OTHER LLMS

We further examine whether the observed performance improvements of our proposed methods translate to LLMs other than Gemini 1.5 Pro. We test the performance of GRASE on both commercial and open-source LLMs, including GPT-4-Turbo (Achiam et al., 2023), Claude-3.0-Opus (Anthropic, 2024), and LLama 3.1 (Dubey et al., 2024) with different parameter sizes. For GPT-4 and Claude, we use their official API service. For Llama 3.1, we use the instruct-turbo version provided by Together AI (Together AI Team, 2024). More details are in Section A.2.

As shown in Figure 4, we observe that GRASE generally helps the models achieve improved performance on PDDL planning over the random exemplar selection (*Random*). With GRASE, models also show improved performance with the increasing number of exemplars, while the improvement can be unstable with random selection. On the other hand, we can observe that, with the help of GRASE, open-source Llama-3.1-70B can achieve similar performance to GPT-4-Turbo with random exemplar selection, which further signifies the usability of GRASE in general application.

## 3.4 PERFORMANCE ON NATURAL LANGUAGE TRIP PLANNING

In the previous sections, we show the significant performance of GRASE-DC on PDDL tasks. We further experiment on Natural Plan (Zheng et al., 2024), where problems, plans, and actions are written in natural language. We use the Trip Planning dataset, where the actions are flights between two cities, which are eventually linked to a full plan to help the travelers cover each city within the given travel and visit time frame. We use the rule-based validator from the original work to parse and evaluate the correctness of the output plans [4].

---

[3] The case that minor errors can lead to total failure adds to the complexity of planning tasks. For example, in *Blocksworld*, the model can forget to put down a block in hand and continue to operate. Exemplars can help remind models to keep hands empty before conducting the next action.

[4] Trip Planning has a natural definition of actions. We apply AS on Calendar Planning with an extended definition of actions in Appendix A.6.

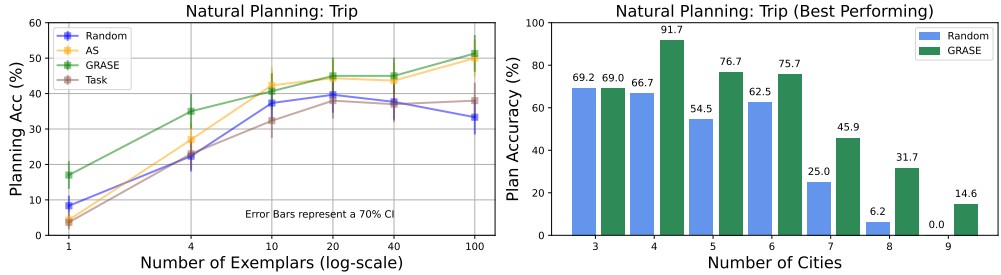

Figure 5: Natural language planning performance on Trip Planning with Gemini 1.5 Pro. (left) ICL performance with different numbers of exemplars and signals for exemplars sampling. Baseline$_{AS}$ (denoted as AS in the figure) denotes the use of Oracle test plans. (right) ICL performance with different problem complexity (denoted by number of cities). *Best Performing* denotes the use of 40 and 100 exemplars for Random and GRASE, respectively.

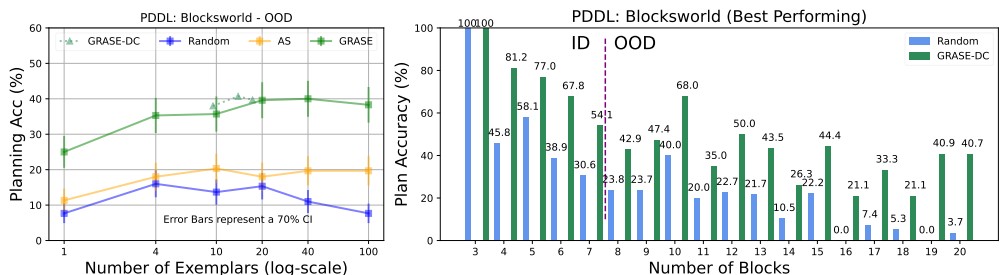

Figure 6: (left) PDDL planning performance on Blocksworld - OOD setting . Unlike Figure 3 (with 3-7 blocks in each example), we test on problems containing 8-20 blocks. Baseline$_{AS}$ denotes the use of Oracle test plans. Numbers of exemplars for GRASE-DC denote the average number of exemplars used over the whole test set with $N_c = 1, 2, 3$; (right) ICL performance with different numbers of blocks. *Best Performing* denotes 4 and 13.9 (on average) exemplars for Random and GRASE-DC, respectively. Since all the exemplar candidates are with 3-7 blocks, we denote test examples with 3-7 blocks as in-distribution (ID) and 8-20 blocks as out-of-distribution (OOD).

As shown in Figure 5 (left), we observe that both Baseline$_{AS}$ and GRASE achieve better performance compared to the random baseline, while sampling exemplars with problem/task similarity do not show significant improvement. Similar to PDDL tasks, using model-generated initial plans to conduct GRASE achieves better results than Baseline$_{AS}$. One reason can be that the reference Oracle plans may not always align with the model preference, e.g., on which cities to start, which brings extra noise. Figure 5 (right) further validates the source of the gain. It depicts a detailed view of the best-performing entries on Figure 5 (left) across different problem complexity (# of cities), noting that the gain from GRASE is across the problems with different # of cities. The improvement is prominent in harder planning problems including more cities.

## 3.5 ANALYSIS: OUT-OF-DISTRIBUTION GENERALIZATION

Motivated by (Bohnet et al., 2024), we note one crucial factor that decides the hardness of the planning problems: the size of the object set. The bigger the size, the harder the planning problem. For Blocksworld, the size denotes the number of the blocks in the problem, i.e. how many blocks are on the table, where some of them are required to be rearranged to reach the goal state. Here, we investigate how our proposed method performs in settings where the size of each test example is different from all the exemplar candidates. This essentially captures the out-of-distribution (OOD) scenario. For the OOD setting, we use exemplar candidates with size 3-7 to solve test examples with size 8-20. From Figure 6 (left), we observe that similar to our findings in Section 3.2, AS achieves significant performance compared to random sampling across various numbers of exemplars, with no decrease in performance when there is a large number of exemplars. We also observe that utilizing the model-generated plans achieves even improved performance than Oracle test plans (Baseline$_{AS}$), which further validates our assumptions in Section 3.2 that GRASE helps models refine its preferred direction of plans with targeted exemplars.

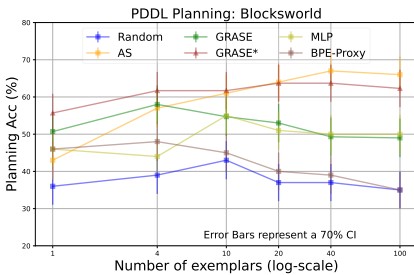 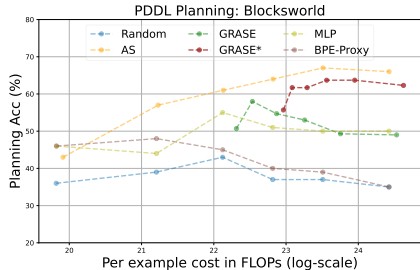

Figure 7: PDDL planning performance on Blocksworld with different ways to empirically approximate AS from the view of (left) numbers of exemplars (right) computation efficiency. GRASE* denotes the iterative application of GRASE only.

As expected, GRASE-DC further improves the model performance by reducing potential noise, so that models can achieve better performance with fewer exemplars, compared to GRASE alone. From the per block size performance in Figure 6 (right), we note that *Random* can completely fail to help models solve hard problems (e.g., problems with 16 and 19 blocks), while GRASE-DC helps achieve consistent gain, which is more significant when the problem is harder: 23.3 points absolute accuracy improvement for out-of-distribution (8-20 blocks) test cases versus 16.3 for in-distribution (3-7 blocks) cases on average.

## 3.6 ANALYSIS: HOW TO APPROXIMATE AS, CONSIDERING THE EFFICIENCY TRADE-OFFS?

In our main experiments, we demonstrate how the plan similarity (AS) is a good source of signal for exemplar selection, as well as how GRASE approximates AS and boosts model performance. It is worth noting that there is extra computing needed for the selection of exemplars compared to *Random*. For Baseline$_{AS}$, computing LCAS is a dynamic programming problem, which can be done in parallel by CPUs in a neglectable time. GRASE requires **one** additional prompt to the LLMs, which is much less than search-based algorithms such as MCTS and ToT, which typically require multiple simulations and one prompt on each node in a tree structure. To achieve further efficiency, we propose additional methods to approximate AS.

**Multi-Layer Perceptron (MLP):** We utilize the Gecko model (Lee et al., 2024) to acquire embeddings, denoted as $m$, to represent the task descriptions and plans. Following our definition in Section 2.1, for an exemplar candidate and a test example, we treat the approximation as a regression task with mean square error (MSE) between $MLP(m(c_d), m(c_p), m(t_d))$ and $Sim_{AS}$ of $\mathcal{A}_c, \mathcal{A}_t$. During training, we sample pairs from candidates. During inference, we use $t_d$ to score and rank all candidates to conduct ICL. The implementation details are in Appendix A.2.

**Byte Pair Encoding as the Proxy**: the above MLP-based method relieves the need for one extra prompt step to acquire $p'$ with an embedding step from a comparative lightweight language model. However, the cost still grows linearly with the number of exemplar candidates. To further improve the efficiency, we propose to add a proxy between the exemplar candidates and the test examples.

Similar to the common practice on tokenization in NLP (Zouhar et al., 2023), we treat actions as *characters* in natural languages and their sequences (i.e., plans) as *sentences*. We conduct Byte Pair Encoding (BPE, Gage, 1994) over the action sequences of the exemplar set to acquire the most commonly appearing sequences, which can be considered as the common subroutines of plans. We denote these proxy action sequences as $b_1, b_2, ..., b_n$ ($n$ is typically 200), which are also action sequences but not necessarily full plans. Before inference, we first pre-compute the $Sim_{AS}^x$ between each $b_x$ and exemplar candidate action sequence $\mathcal{A}_c$. Then during inference, for each $t_d$ of a test example, $Sim_{AS}$ is computed as the weighted average of each proxy similarity: $Sim_{AS} = \sum_{x \in n} \cos \angle(m(t_d), m(b_x)) \cdot Sim_{AS}^x$, where $m$ is the embedding model (e.g., Gecko).

To compare the efficiency of the methods described above, we estimate the floating-point operations per second (FLOPs) needed for exemplar selection and plan generation for each test example. We compute the FLOPs of LLMs following PaLM (Chowdhery et al., 2022), where the number of FLOPs per token is approximately equal to the number of parameters. We assume the use of 1000 candidates (200 natural language tokens per candidate) in the pool, LLama 3.1 405B for inference, and Gecko (1B, 768 dimensions) for embedding. We also assume that Gecko embeddings for the exemplar candidates and BPE tokens are pre-computed (extended details are in Appendix A.3).

From Figure 7, we can observe that, at the best-performing entries, MLP achieves 95% performance of GRASE with around 66% FLOPs and BPE-Proxy achieves 83% performance of GRASE with around 27% FLOPs. Although MLP and BPE-Proxy can not get performance improvement through iteration, they present to be good alternatives of GRASE when there is a limited budget.

## 4 RELATED WORK

**LLM Planning.** Recent investigation on LLM capability (Hao et al., 2023; Valmeekam et al., 2023b; Kambhampati et al., 2024) shows that models can struggle with solving planning tasks directly given the problem descriptions. To comprehensively study this problem, researchers have designed various benchmarks and environments (Valmeekam et al., 2024; Xie et al., 2024; Bohnet et al., 2024; Zheng et al., 2024; Hu & Shu, 2023). In response, there is a line of work discovering the methods to improve planning with LLMs: e.g., applying advanced prompting methods (Silver et al., 2024), utilizing external tools (Hirsch et al., 2024; Hao et al., 2024), or leveraging search-based algorithms (Hao et al., 2023; Lehnert et al., 2024; Zhi-Xuan et al., 2024). In our work, we instruct the model with common and direct in-context learning, without format conversion or customized tool use. We show that simple exemplar selection helps achieve good performance on various planning tasks at a low cost.

**LLM In-Context Learning.** Prompting and in-context learning (ICL) have been considered as the prominent way to interact with LLMs, which achieves significant performance on various tasks (Brown et al., 2020; Wei et al., 2022; Agarwal et al., 2024). Strategical exemplar selection is key to ICL success, where the model performance can be sensitive to perturbations over the exemplars (Zhang et al., 2022). Previous work on exemplar (i.e., demonstration) selection mainly focuses on improving the modeling of task-side similarity (Rubin et al., 2022; Ye et al., 2023). Recently, Auto-CoT (Zhang et al., 2023) discusses how to select exemplars with a fixed cluster on test examples and their rationales. In our work, we demonstrate the effectiveness of the similarity from another perspective, the expected outputs. We show how the action sequence works analytically and empirically on both performance and efficiency (e.g., -DC reduces the exemplar needed). We anticipate the design to be extended to tasks requiring long and dependent sequences as answers, such as web agent trajectory, coding, and math.

**Iterative Refinement with LLMs.** Leveraging model-generated signals to improve model performance has demonstrated its effectiveness in various domains, with supervised fine-tuning (Schick & Schütze, 2021; Welleck et al., 2022; Peng et al., 2023; Chen et al., 2024), prompting (Zelikman et al., 2022), and reinforcement learning (Brooks et al., 2024; Madaan et al., 2024). iPET (Schick & Schütze, 2021) iteratively applies model classification output to improve few-shot learning. STaR (Zelikman et al., 2022) uses the model-generated rationales to improve model reasoning capability. Recently, Self-Refine (Madaan et al., 2024) shows how to utilize feedback on the generation sampled from the same model to refine the model output. Unlike commonsense reasoning tasks such as WSC (Levesque et al., 2012), where the answer is a single choice with a rough overall explanation. In planning tasks, besides the overall goal achievement, the validity of every single action in the sequence is also vital. In our work, we utilize the model-generated plans to re-sample the exemplars in ICL with action sequence similarity. Maintaining the ICL pipeline also allows direct evaluation and easy bootstrapping.

## 5 CONCLUSION

In this paper, we propose GRASE-DC to improve the model planning capability through in-context learning. To do so, we found action sequence similarity (AS) acts as a significant and consistent signal in sampling and filtering exemplars. Following this finding, our GRASE-DC first re-samples high AS exemplars and then curates the selected exemplars with dynamic clustering on AS to achieve a balance of relevance and diversity. Experiments on various planning tasks and settings validate consistent and robust performance improvement with the proposed methods. Extensive analysis demonstrates that our proposed methods deliver these strong results across different LLMs, and even in out-of-distribution scenarios. Given the efficiency and applicability of GRASE-DC, we plan to investigate its application in more complex scenarios and generalization to different environments.

## 6 Acknowledgment

The authors thank Sherry Tongshuang Wu, Hanjun Dai, Bo Dai, Bethany Yixin Wang, Katayoon Goshvadi, Ken Liu, Jiannan Xiang, Xiang Ji, Kaixuan Huang, Xiangyu Qi, Xiang Li, Shikhar Murty, and Vijay Viswanathan for their valuable feedback, and anonymous reviewers for helpful discussions and comments. The authors also thank Google DeepMind teams at large for the insightful discussions and for providing a supportive research environment. At CMU, Xinran Zhao is supported by the ONR Award N000142312840.

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

# A APPENDIX

## A.1 STATISTICS, EXAMPLES, AND PROMPT TEMPLATES

**Statistics.** With the benchmark from Valmeekam et al. (2023a); Bohnet et al. (2024), we construct the exemplar candidate pool with 28,000 examples for Blocksworld, 8,400 examples for Minigrid, 82,000 examples for Logistics, and 2,400 examples for Tetris, which covers the cases with either a small or large number of exemplar candidates. Each example typically contains 400 tokens on average, which varies with the complexity of the problems. For natural language planning, we follow Zheng et al. (2024) to use 1,300 candidate exemplars to conduct trip planning. Please refer to the original papers for further analysis of the statistics. You can also refer to Höller et al. (2020) for extra information about the relevant PDDL domains and corresponding problem generation.

**Examples.** Each example written in PDDL contains a task description that describes the initial conditions, e.g., block b1 is on b2, and goals, e.g., block b2 is on b1. For the exemplar candidate, there is a reference plan that shows one potential way to operate from the initial condition to the goal, e.g., unstack b1 from b2, put b1 on the table, and stack b2 on b1. All the descriptions, goals, and plans are written in a predicate format that can be parsed by PDDL VAL. For natural language planning, descriptions and plans are written in English.

We present the action space and an example for each task as follows:

Blocksworld: (actions: pick-up, put-down, stack, unstack)
TASK: (define (problem BW-rand-4) (:domain blocksworld-4ops) (:objects b3 b2 b1 b4) (:init (on b4 b1)
(clear b4) (clear b2) (on b2 b3) (handempty) (ontable b3) (ontable b1) )
GOAL:(:goal (and (on b2 b3) (on b1 b2) (on b4 b1) )) )
PLAN: (unstack b4 b1) (put-down b4) (pick-up b1) (stack b1 b2) (pick-up b4) (stack b4 b1)

Minigrid: (actions: unlock, move, pickup, pickup-and-loose)
TASK: (define (problem grid_room2) (:domain grid) (:objects p0 p1 p2 p3 ) (:init ; Object types (place p0)
(place p1) (place p2) (place p3) ; Open/locked cells (open p0) (open p1) (open p2) (open p3) ; Connected
cells (conn p0 p1) (conn p0 p2) (conn p1 p0) (conn p1 p3) (conn p2 p0) (conn p2 p3) (conn p3 p2) (conn
p3 p1) ; Lock and key shapes ; Key placement ; Robot placement (at-robot p3) (arm-empty) )
GOAL: (:goal (at-robot p2))
PLAN: (move p3 p2)

Logistics: (actions: load-truck, load-airplane, unload-truck, unload-airplane, drive-truck, fly-airplane)
TASK: (define (problem logistics-c1-s2-p1-a1) (:domain logistics-strips) (:objects a0 c0 t0 l0-0 l0-1
p0 ) (:init (AIRPLANE a0) (CITY c0) (TRUCK t0) (LOCATION l0-0) (in-city l0-0 c0) (LOCATION
l0-1) (in-city l0-1 c0) (AIRPORT l0-0) (OBJ p0) (at t0 l0-0) (at p0 l0-0) (at a0 l0-0))
GOAL: (:goal(and(at p0 l0-1))))
PLAN: (load-truck p0 t0 l0-0) (drive-truck t0 l0-0 l0-1 c0) (unload-truck p0 t0 l0-1)

Tetris: (actions: move-square, move-two, move-l-left/right/up/down
INPUT (define (problem Tetris-4-4) (:domain tetris) (:objects f0-0f f0-1f f0-2f f0-3f f1-0f f1-1f f1-2f f1-3f
f2-0f f2-1f f2-2f f2-3f f3-0f f3-1f f3-2f f3-3f - position nothing- one_square nada- two_straight rightl0
rightl1 - right_l ) (:init (connected f0-0f f0-1f) (connected f0-1f f0-0f) (connected f0-1f f0-2f) (connected
f0-2f f0-1f) ... (connected f3-1f f2-1f) (connected f2-2f f3-2f) (connected f3-2f f2-2f) (connected f2-3f
f3-3f) (connected f3-3f f2-3f) (clear f0-0f) (clear f0-1f) (clear f0-2f) (clear f0-3f) (clear f1-1f) (clear f1-3f)
(clear f3-0f) (clear f3-1f) (clear f3-2f) (clear f3-3f) (at_right_l rightl0 f1-2f f2-2f f2-3f) (at_right_l rightl1
f1-0f f2-0f f2-1f) )
GOAL:(:goal (and (clear f0-0f) (clear f0-1f) (clear f0-2f) (clear f0-3f) (clear f1-0f) (clear f1-1f) (clear
f1-2f) (clear f1-3f) ) ) )
PLAN: (move_l_down f1-0f f2-0f f2-1f f3-0f f3-1f rightl1) (move_l_down f1-2f f2-2f f2-3f f3-2f f3-3f
rightl0)

| Representation | Token | Sentence-T5-base | Sentence-T5-large | Gecko |
|---|---|---|---|---|
| Correlation | 0.17 | 0.25 | 0.25 | 0.29 |

Table 1: Pearson correlation between task similarity and plan similarity (i.e., AS) with different ways to represent the tasks. *Token* denotes capturing the similarity between task descriptions with token overlap.

> Trip planning:
> Task and Goal: You plan to visit 3 European cities for 14 days in total. You only take direct flights to commute between cities. You would like to visit Florence for 6 days. You want to meet a friend in Florence between day 9 and day 14. You would like to visit Barcelona for 5 days. You would like to visit Helsinki for 5 days. Here are the cities that have direct flights: Barcelona and Florence, Helsinki and Barcelona. Find a trip plan of visiting the cities for 14 days by taking direct flights to commute between them.
> Plan: Here is the trip plan for visiting the 3 European cities for 14 days:
> **Day 1-5:** Arriving in Helsinki and visit Helsinki for 5 days.
> **Day 5:** Fly from Helsinki to Barcelona.
> **Day 5-9:** Visit Barcelona for 5 days.
> **Day 9:** Fly from Barcelona to Florence.
> **Day 9-14:** Visit Florence for 6 days.

**Prompt Templates:** We use the simplest prompt to show clearly present the effect of exemplars in ICL. In detail, our prompt is ("Please solve the problem:{task}; Your plan as plain text without formatting:{plan}; done."). For exemplars, we fill in the *task* and *plan* and put them into the context. For text examples, we fill in the *task* and remove the parts after ...*without formatting:*. During writing the exemplars in the context, we start with exemplars with the highest similarity. In our pilot study, the order of starting with the highest or the lowest will not have a huge impact on the performance.

## A.2 IMPLEMENTATION DETAILS

**Metric.** The detailed computation of QA-F1 is as follows: denoting a pair of tasks $t_1$ (test example) and $t_2$ (exemplar candidate) with uni-gram tokens of each task pairs as $\mathcal{T}_{t_1}$ and $\mathcal{T}_{t_2}$, the QA-F1 score can then be written as QA-F1 $= |(\mathcal{T}_{t_1} \cap \mathcal{T}_{t_2})|/|(\mathcal{T}_{t_1} \cup \mathcal{T}_{t_2})|$.

**ICL with LLMs.** For LLMs, we use the API systems, as mentioned in Section 3.3. Due to the instability of API with long context, we test planning performance with few-shot exemplars (1,4,10). In this scenario (DC typically selects 10+ exemplars), diversity is not the major consideration, so we start with conducting a pilot study on the proposed GRASE. The authors would like to note that comparison across commercial API systems can be unfair since the design and inference details are not disclosed.

If applicable, we set the max output token to be 1,600. We also tested our model performance with backbone LLMs with less than 70B parameters. We conduct our experiments on a machine with 8 Nvidia A6000 (40G) GPUs with CUDA 12 installed with inference structure built upon vLLM (Kwon et al., 2023). However, we do not observe performance over 3% planning accuracy on Blocksworld with small-sized models. In these cases, ICL only helps improve the plan validity (i.e., the actions are allowed in the current states).

**MLP.** As a sanity check, we first experiment on how much Gecko understands planning. We compare the Pearson correlation between task similarity and plan similarity (AS) across different ways to capture the similarity between 100 test examples and 400 exemplar candidates. From Table 1, we can observe that using cosine similarity with task description embeddings from Sentence-T5 (Ni et al., 2022) shows a 0.25 correlation coefficient, which surpasses token overlap. Gecko (Lee et al., 2024) achieves a 0.29 correlation, which suggests that Gecko has a good understanding of the potential plans that are relevant to the task descriptions beyond the token level. We also note one

direction to explore in the future: LLM embeddings may achieve higher correlation or better MLP performance with increased cost in the trade-off compared to Gecko.

For the MLP in the main paper, we initialize the network with 2 hidden layers with 400 neurons per layer. We use Adam as the optimizer with a learning rate 1e-5. The whole training process is not sensitive to hyperparameter settings in our experiments. We train our MLP for Blocksworld with 400 exemplar candidates, which leads to 400 x 400 pairs as data points.

**BPE-Proxy.** We wrote a standard Python program to conduct BPE. We conduct the action-level character merging for 500 iterations. We cut off the learned tokens (short action sequences) that appear less than 200 times. In total, we collect 220 tokens to compute the $\text{Sim}_{AS}$ with all exemplar candidates before conducting inference with test examples.

## A.3 DETAILS ABOUT FLOPs COMPARISON

In the main paper, we compare the efficiency of the proposed methods with estimated floating-point operations per second (FLOPs) needed for exemplar selection and plan generation for each test example. Here we provide details about the statistics presented. In general, there are three stages of computation: preparation, exemplar selection, and inference. We denote that there are on average $k$ tokens per task description/plans for either exemplar candidate or test example ($k$ is around 200). Similarly, we assume the use of $N$ candidates, LLama 3.1 (405B) for inference, and Gecko (1B) for embeddings with dimension $d$ ($d$=768).

Then, during preparation, MLP requires processing the embeddings for all exemplar candidates (both task description and plans) once, which requires $2N * k * 1B$ FLOPs. For BPE-Proxy, the computation of the frequent tokens (each is an action sequence) can be neglected. Similarly, it requires processing the embeddings of the frequent tokens, which requires $200 * k * 1B$ FLOPs. Since $N$ is typically larger than 1,000, BPE-Proxy achieves fast preparation compared to MLP. Baseline$_{AS}$ and GRASE require no preparation before observing the test example. For BPE-Proxy, we additionally pre-compute and store the LCAS between frequent tokens and original candidates, which costs $200 * N * k^2$.

During exemplar selection, MLP and BPE-Proxy both require acquiring the embedding of the task description of the test example, which takes $k * 1B$ FLOPs. The task description embedding will be compared with all candidates/tokens for MLP/BPE-Proxy, which requires $N * d$ / $200 * d$ FLOPs for cosine similarity, respectively. GRASE requires one additional prompt, with typically 10 randomly selected exemplars in the context, the FLOPs will then be $(10 + 1) * k * 405B$. Baseline$_{AS}$ and GRASE require the computation of LCAS, which costs $N * k^2$, which can be done by CPU in parallel and neglected compared to the scales of other methods. For GRASE*, it requires double FLOPs compared to GRASE since it requires another round of LLM prompting.

Finally, during inference, with $|D|$ exemplars selected, all methods require $|D| * k * 405B$ FLOPs for inference. In the original paper, we compare methods with the sum of the exemplar selection and the inference cost, since the preparation is only done once before streaming the test examples.

## A.4 MAIN RESULTS ON OTHER PDDL PLANNING TASKS (FULL).

We show the full performance of both GRASE-DC and the iterated version (GRASE-DC*) without zoom-in in Figure 8 (lower row). Similar to our findings in the main paper, we can observe significant and consistent performance improvement with varying numbers of exemplars with GRASE-DC and its iteration.

## A.5 GRASE*: ITERATING GRASE ONLY

In Section 3.2, we present how we leverage GRASE-DC to improve LLM performance on various planning tasks with ICL. We show that the iterative application of GRASE-DC (GRASE-DC*) leads to significant performance improvement with fewer exemplars required. As described in Section 2.2.1, we can also iteratively apply the GRASE step only to have an overview of the performance with a different number of exemplars and make an easy comparison between GRASE and other baselines.

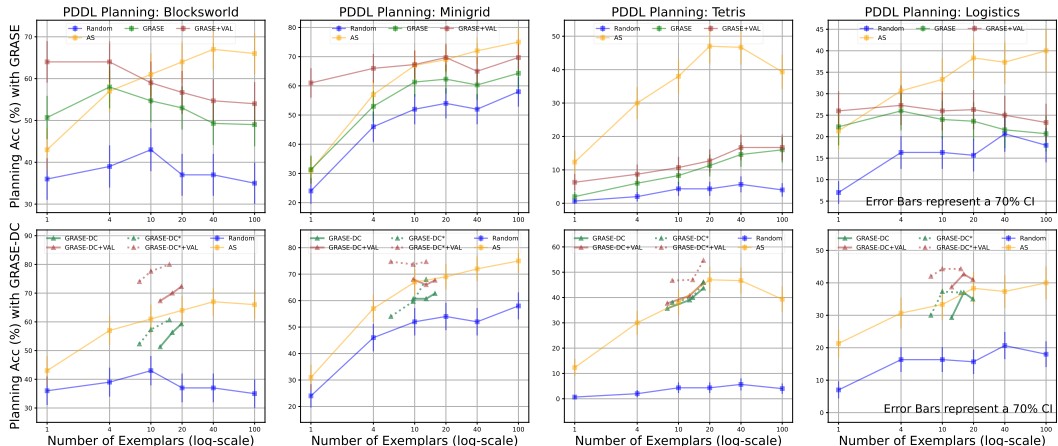

Figure 8: PDDL planning accuracy on various tasks with Gemini 1.5 Pro. Baseline$_{AS}$ denotes the use of Oracle test plans to compute the similarity between test examples and exemplars. Baseline$_{AS}$ and *Random* lines are the anchors across rows. GRASE denotes the use of model output plans from random exemplars. VAL denotes the use of the plan validator. GRASE-DC$^*$ denotes the performance after applying GRASE-DC for another iteration with the model outputs from GRASE-DC.

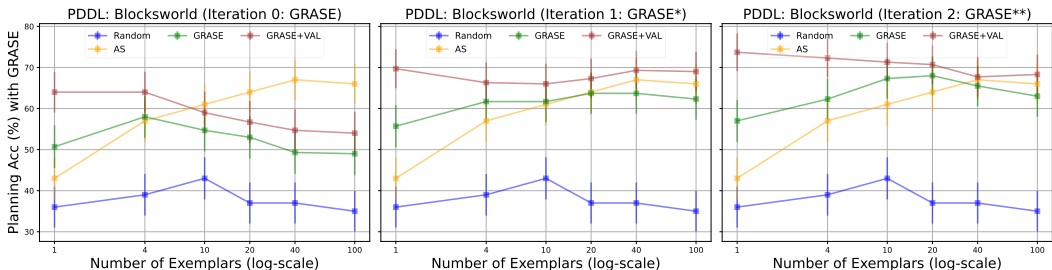

Figure 9: PDDL Planning on Blocksworld with different iterations of GRASE and numbers of exemplars. The *Random* and Baseline$_{AS}$ lines are the same as Figure 3. Error bars in each figure represent a 70% confidence interval. VAL denotes the use of the plan validator and only iterating over the test examples with wrong plans. Iteration 1,2 (*, **) denote the performance after (iteratively) applying GRASE for 1, 2 times, respectively.

As shown in Figure 9, we can observe that, through iteration, GRASE achieves significant performance improvement compared to *Random* across different numbers of exemplars. Each iteration offers an additional ~5 points for absolute planning accuracy improvement. In the second iteration (prompt the model four times in total), GRASE achieves better overall performance compared to Baseline$_{AS}$ that utilizes the Oracle test plans. When GRASE is based on model-generated plans with high accuracy, we can observe less decrease in gains with an increasing number of exemplars.

## A.6 PERFORMANCE ON NATURAL PLAN (EXTENDED)

Calendar planning:
Task and Goal: You need to schedule a meeting for Harold and Patrick for half an hour between work hours of 9:00 to 17:00 on Monday.
Harold's calendar is wide open the entire day. Patrick is busy on Monday during 9:00 to 9:30, 10:30 to 12:00, 12:30 to 13:30, 14:00 to 14:30, 15:00 to 16:30; Find a time that works for everyone's schedule and constraints.
Solution: Here is the proposed time: Monday, 9:30 - 10:00

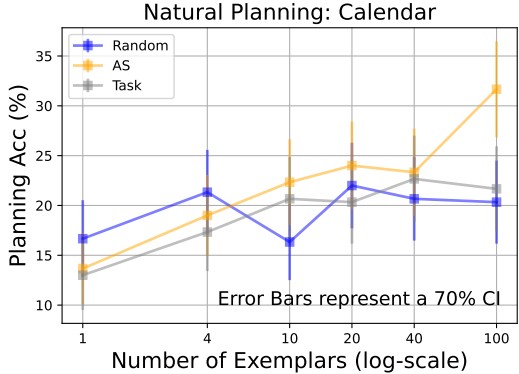

Figure 10: Natural language planning performance on Calendar Planning with Gemini 1.5 Pro with different numbers of exemplars and signals for exemplars sampling. Baseline$_{AS}$ denotes computing action sequence similarity with Oracle test plans.

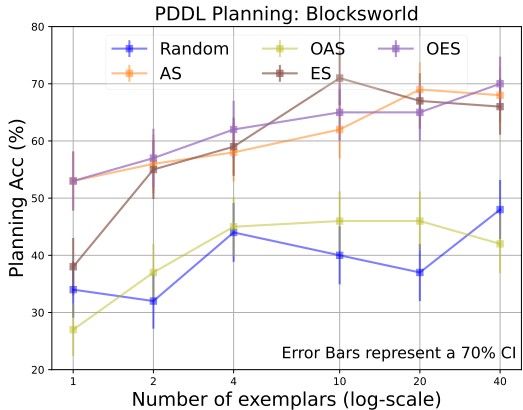

Figure 11: PDDL planning performance on Blocksworld (100 test examples) with different representation methods of plans, where all methods use the Oracle test plans for analysis.

In Section 3.4, we present the performance improvement with AS and GRASE on trip planning. In this section, we further extend our pipeline for other planning tasks in natural language. We notice that the calendar planning task in Natural Plan (Zheng et al., 2024) has a different feature with common planning problems, where the output is not a plan with various actions, but a time slot that is available for all the participants of a meeting. The calendar planning problem can be considered as a constraint-following problem over the given slots of a week (as shown above).

We further implement an elastic AS to test if AS also helps capture constraints in the task description. Similar to the vanilla AS, we represent the constraint as "actions" of each participant with all the available slots in a day with the half an hour interval, e.g, if the available time is 10:30 to 12:00, the corresponding action sequence will be {10:30, 11:00, 11:30}. Except for the changed definition of actions, the whole pipeline remains unchanged.

As shown in Figure 10, we can observe that similar to our findings on PDDL planning tasks and trip planning, Task similarity at the token level shows similar performance as random exemplar selection. Baseline$_{AS}$ performs well when there is a sufficiently large number of exemplars selected, which validates the extensibility of AS on tasks with (ordered) temporal or logical constraints.

## A.7 HOW TO REPRESENT A PLAN, CONSIDERING THE AFFORDANCE AND GRANULARITY?

In our main experiments, we show how representing plans as action sequences (AS) can show great capability in sampling the exemplars. However, AS is not equivalent to the original plan nor uses

| Method/# Exemplars | 1 | 4 | 10 | 20 | 40 | 100 |
|---|---|---|---|---|---|---|
| Random | 36 | 39 | 43 | 37 | 37 | 35 |
| GRASE | 51 | 58 | 55 | 53 | 50 | 49 |
| Random-RS | 52 | 53 | 54 | 52 | 49 | 49 |
| GRASE-RS | 64 | 64 | 59 | 57 | 55 | 54 |
| Random-RS$^*$ | 59 | 59 | 59 | 49 | 49 | 54 |
| GRASE-RS$^*$ | 70 | 66 | 66 | 67 | 69 | 69 |

Table 2: The comparison of planning accuracy (%) between rejection sampling (-RS) with random sampling (Random) and GRASE on Blocksworld. RS$^*$ denotes the iterative application of RS.

leveraging the information of the actual context (i.e., domains in PDDL). For example, consider two steps in the plan: *unstack block A from block B, put-down block A*. AS will represent the plans as *unstack, put down*, with no context of the blocks (e.g., which block is operated?) or the overall situation (e.g., if hands are empty at this step).

To investigate this problem, we first extend AS by adding object affordance information: which object does the action apply to? We first propose to track the AS for each object. When computing the similarity, we consider the average of max AS-based similarly for each object. We denote the use of this similarity to rank exemplars as Object-centric Action Sequences (OAS). For each object, we consider its affordance over the specific action if the action is relevant to multiple objects, e.g., for *unstack block A from block B*, for the AS of block A, we add *unstack_0*, for B, we add *unstack_1* to distinguish the affordance with the order of appearance.

Besides the object relevance, another perspective is the granularity of the actions. For PDDL tasks, VAL can help acquire the actual change of the states with the understanding of the model. For example, when executing *put-down block A*, this action results in the following changes in the state: *delete block A in hand, add hand empty, add block A on the table*, which is a close view on the changes in the problem. We extend our definition of action to execution, i.e., each atomic change on the states. We denote the use of this similarity to rank exemplars as Execution-based action Sequences (ES). Similarly, we also consider Object-centric Execution-based action Sequences (OES) as one kind of representation of plans. A detailed example is as follows:

Given a PDDL plan:{(unstack b1 b6), (put-down b1), (unstack b3 b4), (put-down b3)...}, the action sequence of the plan will be {unstack, put-down, unstack, put-down...}, which ignores the objects as well as their affordance. For Object-centric Action Sequence Similarity (OAS), we first initialize an action sequence for each object, e.g., {b1: unstack-0, put-down, b6: unstack-1 ...}. During comparing two plans written in an object-centric action sequence, for the action sequence of each object in the test example, we find the object with the highest AS in the exemplar candidate. Then we use the macro-average across blocks as the overall OAS similarity.

For Execution-based action sequence Similarity (ES), we maintain the same overall structure but decompose the actions with their actual execution operations on the current states, for example, *unstack* is rewritten to {delete on, delete clear, delete handempty}. Object-centric Execution-based action sequence Similarity (OES) is a combination of OAS and ES.

We further conduct pioneer experiments by comparing these different methods to represent a plan with the first 100 examples of the test set. As shown in Figure 11, we can observe that: except OAS, all methods conducting ICL with plan similarity achieve significantly better performance than random exemplar selection (+15-20 absolute points for the planning accuracy). One reason can be that the action sequence of each object is short, which makes the planning similarity computation unstable. Representing the plans in a detailed way (ES and OES) achieves overall improved performance compared to AS. Above results motivate future work investigating what granularity or if mixed granularity can achieve improved performance over the current pipeline. In our work, to keep the potential generalizability of AS beyond PDDL tasks, we us AS as the representation of a plan in our main experiments.

| Method/# Exemplars | 1 | 4 | 10 | 20 | 40 | 100 |
|---|---|---|---|---|---|---|
| Random | 36 | 39 | 43 | 37 | 37 | 35 |
| Gecko | 35 | 46 | 41 | 44 | 44 | 41 |
| GRASE | 51 | 58 | 55 | 53 | 50 | 49 |

Table 3: The comparison of planning accuracy (%) between embedding-based selection (Gecko) and GRASE on Blocksworld. We use one of the state-of-the-art embedding methods, Gecko (Lee et al., 2024), to compute the similarity of the problem similarity of test questions and exemplars.

## A.8 REJECTION SAMPLING: RANDOM VS. GRASE

In Section 2.2.1, we describe how +VAL is akin to rejection sampling (RS). In this section, we compare the difference between RS with GRASE or random sampling. As shown in Table 2, we can observe that rejection sampling (assuming we have the gold VAL) helps both random and GRASE. However, GRASE-RS shows a 5-12 point accuracy improvement over Random-RS. On the second iteration, the gap between Rand and GRASE becomes larger, which validates our performance gain.

## A.9 DISCUSSION: LINK TO AGENTIC WORKFLOW

Recently, researchers in the LLM agent community have worked on capturing the similarity of trajectories with embedding models to improve agentic the workflow with memory, e.g., Agent-WorkFlow (Wang et al., 2024) and Synapse (Zheng et al., 2023). Following the intuition, we further compare with a baseline using embedding models to capture the similarity between problems. We use one of the state-of-the-art embedding methods, Gecko (Lee et al., 2024), to compute the similarity of the problem similarity of test questions and exemplars. We then use the similarity to rank and select the exemplars and conduct ICL.

As shown in Table 3, we can observe that GRASE shows consistent and significant performance improvement over using the off-the-shelf embedding models. This observation further suggests the potential application of the proposed GRASE in the agentic workflow by capturing the similarity of trajectories via LCAS.

For the current scope of this paper, we focus on presenting the gains from the novel GRASE on planning tasks to show a clear contribution to planning. Besides PDDL tasks, we believe real-world simulated tasks are another important future direction, e.g., Web Arena (Zhou et al., 2023), Mind2Web (Deng et al., 2023), and ALFWorld (Shridhar et al., 2021). Besides the planning-side capability, in simulated tasks, the LLM-based agents are also additionally required to capture and understand the complex environmental feedback per action (Gu et al., 2024). Applying the state-based plan representation, i.e., ES in Appendix A.7, can potentially make GRASE applicable to exemplar selection in environmental feedback simulation and eventually the whole pipeline to solve simulated tasks. The authors sincerely hope the simple yet effective intuition we showcase can also inspire further ideas in the community in different directions.

### A.10 QUALITATIVE EXAMPLES OF PAIRS WITH HIGH INPUT SIMILARITY VS. PLAN SIMILARITY

To qualitatively illustrate the differences between exemplars with high task similarity and plan similarity, we present a test example as follows with the exemplars with the highest task/plan similarity as follows. From the comparison, we can observe that, although plan similarity captures an exemplar with the same number of blocks and similar conditions from the appearance. The required plan can be different and unrelated. On the other hand, despite the different block sizes, the exemplar with high plan similarity is similar to the original problem from the essence: e.g., b5, b4 in the original problem are dealt in a similar way with b3, b2, respectively in the exemplar. As a result, putting the high-task-similarity exemplars in the context results in a wrong plan, while the high-plan-similarity example leads to a correct plan, which is also different from the Oracle test plan.

| **Original** | **High-Task-Similarity** | **High-Plan-Similarity** |
|---|---|---|
| (define (problem BW-rand-6) | (define (problem BW-rand-6) | (define (problem BW-rand-5) |
| (:domain blocksworld-4ops) | (:domain blocksworld-4ops) | (:domain blocksworld-4ops) |
| (:objects b1 b5 b2 b4 b3 b6) | (:objects b5 b4 b6 b3 b2 b1) | (:objects b1 b3 b5 b2 b4) |
| (:init (on b2 b5) | (:init(on b3 b6) | (:init (clear b4) |
| (clear b3) | (handempty) | (on b5 b3) |
| (on b4 b2) | (clear b4) | (on b2 b5) |
| (handempty) | (clear b1) | (handempty) |
| (on b3 b4) | (on b1 b2) | (on b3 b1) |
| (ontable b1) | (on b4 b5) | (on b4 b2) |
| (on b5 b1) | (ontable b5) | (ontable b1)) |
| (ontable b6) | (ontable b6) | (:goal (and |
| (clear b6) ) | (on b2 b3) ) | (on b3 b2) |
| (:goal (and | (:goal (and | (on b1 b3) ))) |
| (on b5 b4) | (on b6 b1) ))) | Plan: |
| (on b3 b5) ))) | PlAN: | (unstack b4 b2) |
| PlAN: | (unstack b1 b2) | (put-down b4) |
| (unstack b3 b4) | (put-down b1) | (unstack b2 b5) |
| (put-down b3) | (unstack b2 b3) | (put-down b2) |
| (unstack b4 b2) | (put-down b2) | (unstack b5 b3) |
| (put-down b4) | (unstack b3 b6) | (put-down b5) |
| (unstack b2 b5) | (put-down b3) | (unstack b3 b1) |
| (put-down b2) | (pick-up b6) | (stack b3 b2) |
| (unstack b5 b1) | (stack b6 b1) | (pick-up b1) |
| (stack b5 b4) | **Gemini Output**: | (stack b1 b3) |
| (pick-up b3) | (unstack b3 b4) | **Gemini Output**: |
| (stack b3 b5) | (put-down b3) | (unstack b4 b2) |
| | (pick-up b3) | (put-down b4) |
| | (stack b3 b5) | (unstack b2 b5) |
| | (pick-up b5); wrong | (put-down b2) |
| | (stack b5 b4) | (unstack b3 b4) |
| | | (stack b3 b2) |
| | | (pick-up b5) |
| | | (stack b5 b4) |
| | | (pick-up b3) |
| | | (stack b3 b5) ; correct |

### A.11 PERFORMANCE SCORES IN A TABLE

In the main paper, we present our results in line charts to capture the trends with varying numbers of exemplars. We further present the performance of baselines and the proposed GRASE-DC in Table 4. From the table, we can observe that GRASE+DC performs well on various tasks. The -DC step helps achieve the best performance under most scenarios, when it is not, it helps achieve comparable performance with fewer exemplars.

| Method | # of Exemplars | | | | | | with -DC ($N_c$) | | | Best |
|---|---|---|---|---|---|---|---|---|---|---|
| | 1 | 4 | 10 | 20 | 40 | 100 | 1 | 2 | 3 | |
| *Blocksworld* | | | | | | | | | | |
| Random | 36 | 39 | 43 | 37 | 37 | 35 | | | | 43 |
| Task | 25 | 36 | 40 | 39 | 41 | 36 | | | | 41 |
| AS | 43 | 57 | 61 | 64 | 67 | 66 | | | | 67 |
| GRASE | 50.7 | 58 | 54.7 | 53 | 49.3 | 49 | 51.3 (6.6) | 56.3 (9.6) | **59.3** (12.2) | 59.3 |
| GRASE+VAL | 64 | 64 | 59 | 56.7 | 54.7 | 54 | 67.3 (6.6) | 70 (9.6) | **72.3** (12.2) | 72.3 |
| GRASE* | 55.7 | 61.7 | 61.7 | **63.7** | **63.7** | 62.3 | 52.3 (6.6) | 57.3 (9.6) | 60.7 (14.7) | 63.7 |
| GRASE*+VAL | 69.7 | 66.3 | 66 | 67.3 | 69.3 | 69 | 74 (6.6) | 77.6 (9.6) | **80** (14.7) | 80 |
| GRASE** | 57 | 62.3 | 67.3 | **68** | 65.5 | 63 | | | | 68 |
| GRASE**+VAL | **73.7** | 72.3 | 71.3 | 70.7 | 67.7 | 68.3 | | | | 73.7 |
| *Minigrid* | | | | | | | | | | |
| Random | 24 | 46 | 52 | 54 | 52 | 58 | | | | 58 |
| Task | 26 | 32 | 37 | 48 | 48 | 49 | | | | 49 |
| AS | 31 | 57 | 67 | 69 | 72 | 75 | | | | 75 |
| GRASE | 31.3 | 53 | 61.3 | 62.3 | 60.3 | **64.3** | 60.7 (9.7) | 60.7 (12.9) | 62.7 (15.7) | 64.3 |
| GRASE+VAL | 61 | 66 | 67.3 | 69.7 | 65 | **69.7** | 68 (9.7) | 66 (12.9) | 67.7 (15.7) | 69.7 |
| GRASE* | | | | | | | 54 (5.9) | 59.7 (9.7) | **68** (12.9) | 68 |
| GRASE*+VAL | | | | | | | **74.7** (5.9) | 73.7 (9.7) | **74.7** (12.9) | 74.7 |
| *Tetris* | | | | | | | | | | |
| Random | 0.7 | 2.0 | 4.3 | 4.3 | 5.7 | 4.0 | | | | 5.7 |
| Task | 0.7 | 4.0 | 5.3 | 7.0 | 6.0 | 5.7 | | | | 7.0 |
| AS | 12.3 | 30.0 | 38.0 | 47.0 | 46.7 | 39.3 | | | | 47.0 |
| GRASE | 2 | 6 | 8.3 | 11.3 | 14.6 | 16 | 35.7 (7.8) | 39 (12.8) | **43.7** (17.4) | 43.7 |
| GRASE+VAL | 6.3 | 8.7 | 10.7 | 12.7 | 16.7 | 16.7 | 37.7 (7.8) | 40.7 (12.8) | **46** (17.4) | 46 |
| GRASE* | | | | | | | 38 (8.7) | 40 (13.7) | **46** (17.4) | 46 |
| GRASE*+VAL | | | | | | | 46.7 (8.7) | 47 (13.7) | **54.7** (17.4) | 54.7 |
| *Logistics* | | | | | | | | | | |
| Random | 7.0 | 16.3 | 16.3 | 15.7 | 20.7 | 18.0 | | | | 20.7 |
| Task | 20.3 | 30.0 | 32.7 | 38.7 | 36.0 | 35.0 | | | | 38.7 |
| AS | 21.3 | 30.7 | 33.3 | 38.3 | 37.3 | 40.0 | | | | 40.0 |
| GRASE | 22.3 | 26 | 24 | 23.6 | 21.6 | 20.7 | 29.3 (12.3) | **37** (16.1) | 35 (19.7) | 37 |
| GRASE+VAL | 26 | 27.3 | 26 | 26.3 | 25 | 23.3 | 38.7 (12.3) | **42.7** (16.1) | 41 (19.7) | 42.7 |
| GRASE* | | | | | | | 30 (7.8) | **37.3** (10) | 37 (15.1) | 37.3 |
| GRASE*+VAL | | | | | | | 42 (7.8) | **44.3** (10) | **44.3** (15.1) | 44.3 |
| *Trip* | | | | | | | | | | |
| Random | 8.3 | 22.3 | 37.3 | 39.7 | 37.7 | 33.3 | | | | 39.7 |
| Task | 3.7 | 23.0 | 32.3 | 38.0 | 37.0 | 38.0 | | | | 38.0 |
| AS | 4.3 | 27.0 | 42.3 | 44.3 | 43.7 | 50.0 | | | | 50.0 |
| GRASE | 17.0 | 35.0 | 40.7 | 45.0 | 45.0 | 51.3 | | | | 51.3 |
| *Blocksworld - OOD* | | | | | | | | | | |
| Random | 7.7 | 16.0 | 13.7 | 15.3 | 11.0 | 7.7 | | | | 16.0 |
| AS | 11.3 | 18.0 | 20.3 | 18.0 | 19.7 | 19.7 | | | | 20.3 |
| GRASE | 25 | 35.3 | 35.7 | 39.6 | 40 | 38.3 | 38 (9.5) | **40.7** (13.9) | 39.7 (17.3) | 40.7 |

Table 4: PDDL Planning Performance in planning accuracy (%) on various tasks with Gemini 1.5 Pro. With -DC the use of dynamic clustering in the pipeline, where changing $N_c$ typically leads to 7-20 exemplars selected. For entries under -DC, numbers in brackets denote the averaged number of exemplars across the test data. ** denotes the iterative application of GRASE on the model-generated plans form GRASE* (Figure 9). Blocksworld - OOD denotes the performance on out-of-distribution problems in Figure 6. Best entries per row for each variant of GRASE or GRASE-DC are in **bold**.

| Abbr. | Term | Description |
|---|---|---|
| Random | Random Selection | We use Random in the figures to denote the baseline with randomly selected exemplars. |
| AS | Action-sequence Similarity | AS denotes the action sequence similarity, which we capture with $Sim_{AS}$ based on LCAS. We use $Baseline_{AS}$ to denote the proposed method of ranking the exemplar candidates with their action sequence similarity with the Oracle test plans. |
| VAL | PDDL Validator | We use VAL to denote the validator of the plans. +VAL in the figures denotes the variant in our pipeline, where we only operate on examples that LLMs failed in the last round. |
| ICL | In-Context Learning | We use ICL to refer to the classic ICL pipeline where we put each exemplar in the templates and sequentially list them as the context for prompting. |
| GRASE | Generative Resampling of Action sequence Similar Exemplars | We propose GRASE as the first stage of our pipeline that utilizes the model-generated plans to help re-sample the exemplars. |
| DC | Dynamic Clustering | We propose DC as the second stage of our pipeline that resamples the exemplars from the GRASE stage by their mutual relevance calculated by AS. |
| * | Iteration | We use * to refer to the iterative application of our pipeline, where the number of * denotes the number of iterations. |
| MLP | Multi-Layer Perceptron | We use MLP to denote the specific MLP we trained to approximate AS between the test example and a candidate exemplar with the exemplar task, exemplar plan, and test example task. |
| BPE-Proxy | Byte Pair Encoding as the Proxy | We use BPE to refer to the specific method to acquire tokens for planning, where each "character" is an action. BPE-Proxy denotes the method to only calculate the similarity between the tokens and the test examples so that the cost at inference time does not scale linearly with an increasing number of exemplar candidates. |

Table 5: Referential explanations to all the terms

A.12 REFERENCE FOR COMMON TERMS

To improve the readability of the paper, we wrote an explanation for a set of important abbreviated terms we used in Table 5.

