# OpenReview forum: "Improving Large Language Model Planning with Action Sequence Similarity"
_ICLR.cc/2025/Conference — ICLR 2025 Poster_

### Official Review · Reviewer_BrrL · 2024-10-23

**Soundness:** 2
**Presentation:** 2
**Contribution:** 2
**Rating:** 5
**Confidence:** 4

**Summary:**

This paper proposes GRASE-DC, a two-stage selection algorithm designed to identify effective and diverse exemplars from candidate plans, thereby enhancing in-context learning effectiveness. In the first stage, GRASE calculates the Action Similarity (AS) between different plans based on their sequence of actions. In the second stage, DC refines the selected exemplars using dynamic clustering based on AS, ensuring a balance between relevance and diversity. The effectiveness of GRASE-DC is demonstrated through extensive experiments, which show significant performance improvements over baseline methods.

**Strengths:**

[+] The overall structure is compact and clear.

[+] The use of four benchmark domains provides convincing results.

[+] The analysis section effectively discusses the efficiency of different methods for calculating Action Similarity (AS).

**Weaknesses:**

[-] My concern is that such methods may be difficult to adopt in real-world applications. I feel that in-context learning is not the optimal solution for improving the planning capabilities of LLMs, especially considering the existing o1. In-context learning is more suitable for quickly adapting the model to output desired formats or providing it with hints to follow, rather than using very long in-context prompts to enhance the model's capabilities.

[-] The paper should at least compare some baseline approaches that use the similarity between plans. In the LLM-agent area, there are papers that employ similar ideas, using a similarity metric to select trajectories from the memory module. Implementing such a simple baseline would not be difficult and could provide a better complement to the experiments, enhancing their convincingness.

**Questions:**

1. How do you justify that only the sequence of actions is important for capturing the plan properties? Have you considered including intermediate states (i.e., the object states and information) in the plan?

2. Could you provide a detailed pseudocode for GRASE-DC? I am a bit confused with the steps in Diversity Clustering.

---

> ### Author Response · Authors · 2024-11-19
> **Thank you for your review.**
>
> Thank you for your insightful and detailed feedback. We address your questions below.
>
> Q1: **ICL with GRASE.**
> Thanks so much for initiating the discussion.
>
> Regarding the real-world applications, we show our generalizability from two perspectives: (1) In Section 3.4 and Appendix A.6, we show performance gain on natural language planning tasks with inputs/outputs in the format of Google Flights and Google Calendar from data in [1] Natural Plan (Zheng et al., 2024); (2) In Section 3.5, we show the out-of-distribution generalization of our method, given that real-world applications can often be more complex than benchmarks. We also observe consistent performance with GRASE.
>
> Regarding ICL, Our method does not contradict existing test-time search algorithms that are potentially used in o1. When making decisions at each node, providing good exemplars sampled by GRASE can also help in collaboration.
>
> On the other hand, our method is like a combination with ICL that provides format and Retrieval Augmented Generation (RAG) that provides knowledge on how to plan correctly. Unlike ICL with a random/fixed set of exemplars with the desired output format, our method selects a specific set of exemplars per test question based on the model-generated plans (with its desired direction of solving the problem). The great performance gain from our proposed method on out-of-distribution data suggests one potential reason why GRASE works: these targeted exemplars provide knowledge that can help models refine their preferred directions.
>
>
> Q2: **Baselines on the plan side (Weakness 2 and Q1).**
> Thanks for the insights. We indeed have compared some baseline approaches that use the similarity between plans in our current paper.
>
> Similar to your insights, we also consider including object states and information in representing the plans. As mentioned in lines 202-203 of the paper, in Section A.7 in the appendix, we present the comparison of different ways to represent a plan: whether to include objects (OAS) or whether to consider the states (ES, OES). As shown in Figure 11, AS empirically performs better than OAS and with similar performance as ES. Yet AS has better potential extensibility than ES. This empirical comparison justifies our choice of AS in the paper.
>
> We denote [2] AgentWorkFlow (Wang et al., 2024) and [3] Synapse (Zheng et al., 2024) as potential papers you refer to in the LLM-agent area. Please specify if there are other papers you refer to. Motivated by Synapse, we propose to incorporate the baseline to select the exemplars with embeddings and cosine similarity.
>
> We use one of the best-performing embedding models, Gecko (Lee et al., 2024), to conduct similar experiments (Gecko demonstrates better retrieval performance than text-embedding-3-large from Open AI). The performance is as follows and we still observe better performance with GRASE on Blocksworld with the experiment setting in our paper:
>
> | Method/# exemplars | 1 | 4 | 10 | 20 | 40 | 100 |
> | :-------------------: | :-: | :-: | :-: | :-: | :-: | :-: |
> | Rand. | 36 | 39 | 43 | 37 | 37 | 35 |
> | Gecko | 35 | 46 | 41 | 44 | 44 | 41 |
> | GRASE | 51 | 58 | 55 | 53 | 50 | 49 |
>
> The authors sincerely hope the plan representation ablation results mentioned in lines 202-203 and presented in Section A.7, together with the new baseline on the recent work from the LLM-agent area, can address your concern regarding plan similarity.

---

> > ### Comment · Reviewer_BrrL · 2024-11-21
> > **Thanks for the thorough response.**
> >
> > Thank you to the authors for the thorough and prompt response. The clarifications and new empirical results have addressed some of my earlier concerns. I have a few additional questions:
> >
> > 1. What is the exact latency overhead of GRASE-DC, and how does it compare to the baseline methods used in your paper? I.e., the extra latency caused by GRASE-DC.
> >
> > 2. In the first step of GRASE, the LLM generates an initial plan based on random exemplars. I assume that, in this case, the LLM is unlikely to produce the correct plan on its first attempt. After this step, GRASE selects more effective exemplars based on AS. Have you considered expanding the first step to include multiple iterations, and gradually refining the plan over several rounds? And how would this affect the subsequent procedures in GRASE-DC and its final performance.

---

> > > ### Author Response · Authors · 2024-11-21
> > > **Thanks for the discussion.**
> > >
> > > Thank you for your detailed feedback and for engaging in the discussion. We are glad that our clarifications and new empirical results have addressed your early concerns. We kindly request that you consider increasing your rating accordingly. We address your additional questions below.
> > >
> > > Q1: **Exact Latency**
> > > In Section 3.6, we compare the method efficiency from the perspective of FLOPs. MLP achieves 95% performance of GRASE (with model-generated plans) with around 66% FLOPs and BPE-Proxy achieves 83% performance of GRASE (with model-generated plans) with around 27% FLOPs.
> > >
> > > With reasonable hardware assumptions, latency is roughly linearly correlated with the amount of computing needed (measured by FLOPs), which suggests that, for example, GRASE with model generated plan takes 1.5 times the averaged latency compared to the MLP version we proposed.
> > >
> > > The exact latency measured in seconds will be largely dependent on the actual infrastructure (GPU and network) and API used. For example, if you are using Claude-Opus (our experiment in Section 3.3), the actual extra latency will be on average 0.38 seconds more per test query with GRASE (originally 0.76 seconds) with reasonable settings and network conditions.
> > >
> > > Q2: **Multiple iterations**
> > > Thanks for the insightful comments. That is exactly what we intend to propose in the paper. In our design of GRASE, the ICL pipeline is maintained with exemplars changed, which makes iteration easy.
> > >
> > > From lines 338 - 345, we introduce the iterated version of GRASE-DC. We show the performance gain from the first iteration (solid lines) to the second (dashed lines) in Figure 3 (lower row). As you anticipate, GRASE-DC∗ (iterated version), either with VAL (brown lines) or without VAL (green lines), helps achieve higher performance with few exemplars (pushing the curve to the upper left), compared to the first attempt.
> > >
> > > Additionally, as mentioned in lines 336-337, in Appendix A.5, we also show the iterative version of GRASE only with more iterations. From Figure 9, we can observe that, with multiple iterations, as expected, we can get improved performance on model planning.
> > >
> > >
> > > We sincerely appreciate your feedback. We hope our responses convince you about the efficiency and performance of our methods. If you have further questions or concerns, please let us know.

---

> > > ### Author Response · Authors · 2024-12-01
> > > **Thanks again for engaging in the discussion...**
> > >
> > > Thank you for your time and engagement in the discussion. We truly appreciate your detailed review and we are glad that our explanation of ICL, baselines motivated from agent trajectory, and implementation details addressed your earlier concerns. We are also pleased to answer your additional questions.
> > >
> > > In our last round of responses, we further discussed the exact latency and the gain from iterations (GRASE-DC* in the original paper). We would greatly appreciate it if you could share any further insights or suggestions. Otherwise, if you feel our additional explanations and experiments have satisfactorily addressed your concerns, we kindly ask: if you can consider increasing your score before the end of the discussion to support our paper's acceptance. Thanks!

---

> ### Author Response · Authors · 2024-11-19
>
> Q3: **Pseudo Code of GRASE-DC.**
> Thanks for asking, the brief pseudo-code of GRASE-DC is as follows (for one test example). Let us know if there are further details needed.
>
> let Nc be the factor controlling -DC in the paper, with per_cluster_sample_max controlling the max number of exemplars from each cluster
>
> Input: test_task, generated_test_plan, candidate_task_list, candidate_plan_list
>
> Function compute_sim_as(as1, as2):
> >return len(LCAS(as1,as2))**2/(len(as1),len(as2)),
>
> Function find_action_sequence_similarity(generated_test_plan, candidate_plan_list):
> >sim_as = [] \
> > for candidate_plan in candidate_plan_list:
> >> this_as = compute_sim_as(generated_test_plan, candidate_plan_list) \
> >> sim_as.add(this_as)
>
> >return sim_as
>
> Function find_exemplars(sim_as, candidate_plan_list):
> >selected_candidates = [] \
> >mean, std = mean(sim_as), std(sim_as) \
> >candidates_to_be_clustered = [candidate_plan_list[i] if sim_as[i] > (mean+std) for i in range(len(candidate_plan_list))] \
> >n_cluster = len(candidates_to_be_clustered)**0.25 * Nc \
> >candidate_pair_sim_as = [compute_sim_as(as1, as2) for pairs in candidates_to_be_clustered] \
> >clusters = AgglomerativeClustering(n_cluster).fit(reciprocal of candidate_pair_sim_as) \
> >cluster_visit_count = {} \
> >for i, candidate in enumerate(candidates_to_be_clustered):
> >>if sim_as[i] > (mean+3std): selected_candidates.add(candidate) # high relevance \
> >>elif cluster_visit_count[clusters.labels[i]] <= per_cluster_sample_max:
> >>>selected_candidates.add(candidate) # pass the diversity requirement \
> >>>cluster_visit_count[clusters.labels[i]]  += 1
>
> >return selected_candidates
>
> Function ICL_for_one_question(test_task, selected_candidates, LLM):
> >new_model_generated_plan = LLM(selected_candidates + test_task) # with templates \
> >return new_model_generated_plan
>
> Finally, we sincerely hope that we have addressed all your concerns in the rebuttal with discussion and the new experiments. We would be happy to discuss if there are any further points.
> Considering that you find our extensive experiments convincing, our analysis effective and our writing clear, we would appreciate it a lot if you could increase your overall assessment score of the paper if there are no additional issues.
>
> **References:**
>
> [1] NATURAL PLAN: Benchmarking LLMs on Natural Language Planning
>
> [2] Agent Workflow Memory
>
> [3] Synapse: Trajectory-as-Exemplar Prompting with Memory for Computer Control
>
> [4] Gecko: Versatile Text Embeddings Distilled from Large Language Models

---

### Official Review · Reviewer_BjER · 2024-10-23

**Soundness:** 3
**Presentation:** 3
**Contribution:** 3
**Rating:** 5
**Confidence:** 4

**Summary:**

This paper explores the action sequence as a key signal to measure the similarity between exemplars in planning tasks. Based on this finding, the paper introduces a two-stage pipeline, GRASE-DC, to select similar in-context examples while maintaining diversity and relevance. Experiments on both PDDL and natural language planning tasks show the promising performance of the proposed exemplar selection strategy. Further analysis includes OOD generalization and some options to replace AS when considering efficiency.

**Strengths:**

- This paper breaks away from traditional task similarity and instead utilizes action sequence similarity to select exemplars for ICL, enhancing the model's planning performance, which is simple yet effective.
- The GRASE-DC shows great generalization performance on more complex tasks.
- Endeavors have been made to pursue the efficiency of the exemplar selection process.

**Weaknesses:**

1. The method of using clustering algorithms to improve the relevance and diversity of the selected exemplars has been proposed by other paper [1] before.
2. The selected evaluation datasets lack some real world simulated tasks such as ALFWorld, Mind2Web, ScienceWorld, etc.
3. Though the VAL mechanism is referenced from other paper, the authors are suggested to introduce it briefly in the paper to enhance the readability as VAL appears frequently in the paper.

[1] Automatic Chain of Thought Prompting in Large Language Models, ICLR 2023

**Questions:**

- Why are MLP and BPE-Proxy more efficient than LCAS, as encoding the action sequence to embeddings itself requires inference time?
- Why is Sim$_{AS}$ designed as shown in line 141, whereas based on experience LCAS($A_i$, $A_j$) should be squared?
- How is the decision made for the iterative version of GRASE-DC on whether to iterate? Since we do not have test data trajectories and answers, based on efficiency, iteration is only needed in case of errors.

---

> ### Author Response · Authors · 2024-11-19
> **Thank you for your review.**
>
> Thank you for your valuable and detailed feedback. We address your questions below.
>
> Q1: **Related work: Auto-CoT.**
> Thanks for mentioning the insightful related work. We will include a discussion on work in the literature.
> One critical difference between our dynamic clustering (-DC) and Auto-CoT is that we construct clusters with a dynamic set of exemplars based on the relevance to **each** test example, instead of clusters on the test questions.
> For -DC, we further explore (1) using the proposed action sequence similarity as a novel plan-side distance signal to conduct Agglomerative Hierarchical Clustering, instead of the embeddings from SentenceBERT in Auto-CoT; (2) utilizing per-test example clusters to help automatically decide the number of exemplars for each test question (empirically, 27.3% fewer exemplars are needed), while in Auto-CoT, there is a fixed number of exemplars/demonstrations (8 per test question in Section 3.3 of the paper) before rule-based pruning.
>
> Q2: **Real world tasks.**
> Thanks so much for the suggestions. In our experiments, to test the real-world generalizability of GRASE, we conduct experiments on natural language planning with outputs from Google Flights and Google Calendar from data in Natural Plan (Zheng et al., 2024). In Section 3.4 and Appendix A.6, we show how GRASE and AS have great performance on these real-world motivated tasks.
>
> The authors agree with the importance of real-world simulated tasks and regard ALFWorld, Mind2Web, etc as important future directions. We will add discussion on this direction, yet for the current scope of our paper, we focus on presenting the gain from the novel GRASE-DC on planning tasks, while simulated tasks often also require model capability in capturing environmental dynamics (e.g., abstract views of the environmental feedback) and tool use (e.g., embodied cleaning-related actions in AFWorld with specific APIs), besides planning. To show a clear contribution to planning, we conduct our experiments on PDDL and natural language planning tasks at this stage.
>
> Q3: **PDDL Validator (VAL).**
> Thanks so much for your suggestion on improving the presentation. We will extend our description in Section 2.1 regarding the details of PDDL validators: rule-based systems that validate the planning success by checking if a given action is viable at the current states given a PDDL domain description and check if the goals are satisfied final states. We will extend our discussion in lines 198-201: for +VAL, we only conduct the second iterations where a PDDL validator outputs that the model-generated plans are wrong.
>
> Q4: **Efficiency.**
> MLP and BPE-proxy are more efficient than GRASE with LCAS since the latter requires one more round of inference on the LLM to get the initial plans, which takes a longer time than the smaller model inference for the embeddings with MLP/BPE-Proxy.
>
> Q5: **LCAS.**
> Thanks so much for pointing out the typo. In the implementation, we used the squared length of the common action sequence to rank the scores. We will correct it in the final version.
>
> Q6: **Whether to Iterative.**
> With the Validator (VAL) in PDDL, while searching for a correct plan when building a non-neural planner, it is common to use VAL to validate if the action/plan is incorrect. If we assume the use of VAL for planning with LLMs, similar to your intuition, we only iterate on the incorrect examples and show great improvement through iteration in Figure 3 in the paper.
>
> If we assume that there is no VAL to decide if a generated plan is wrong, we can iterate until the performance converges, similar to the early stopping mechanism. Since there is no guarantee on the model performance ceiling, due to the cost, we iterate until we hit a given budget in our current paper (2 additional generations with the LLMs).
>
>
> Finally, we appreciate your feedback. We hope our responses convince you about the problem's motivation and our methods. We kindly request that you review our responses and consider increasing your rating accordingly. If you have further questions or concerns, please let us know.

---

> ### Comment · Reviewer_BjER · 2024-11-22
> **Reply to authors' response**
>
> Thank you for providing your feedback. The following issues lead me to maintain my score:
>
> 1. I still have reservations regarding the novelty of the clustering approach.
> 2. I disagree with the authors' assertion. In fact, simulated tasks are exactly crucial "To show a clear contribution to planning." And travel planning tasks also need the LLMs to understand the dynamic environments. The authors have not sufficiently explained why they chose to focus solely on pure planning tasks when it is clear that their method is also applicable to simulated tasks.
> 3. According to ICLR's rebuttal policy, authors are allowed to make modifications to their paper. Why are the authors waiting until the so-called "final version" to incorporate the feedback from the reviewers rather than making changes now?

---

> ### Author Response · Authors · 2024-11-27
> **Sincerely appreciate your engagement in the discussion...**
>
> Sincerely thank you for your detailed feedback and for engaging in the discussion. We are glad that our clarifications have addressed part of your early questions and concerns. We address your additional questions below.
>
> Q1: **Clustering Approach**
> Thank you again for the discussion. The main focus of our paper is to validate and explore how action sequence similarity (AS) can improve LLM planning. With GRASE, we show how AS can empirically rank the exemplars. With -DC, we show AS can be applied to clustering to further improve performance and efficiency (by cutting down the exemplars needed). The methodological contribution is not on the clustering approach (we use the classical Agglomerative Hierarchical Clustering) but on the discovery of (1) how AS can be a signal; and (2) how we utilize the clusters.
>
> As in our last round of discussion, with similar motivation, Auto-CoT utilizes clustering to improve ICL with a fixed cluster on the test questions. In our -DC, we conduct the clustering in an adaptive manner: for each test question, we select the exemplars with good relevance (mean+1std) to construct a specific cluster for this test question as the first filtering step. Next, we prune the exemplar set for this specific question with the cluster and our proposed method described in Section 2.2. We show how this method can empirically reduce the overall exemplar needed and dynamically decide how many exemplars are needed per test question. The source of clustering signals and the point-wise per test question control are how we are different from Auto-CoT. The authors indeed enjoyed reading the Auto-CoT paper and added the discussion to both the -DC section (Section 2.2.2) and related work section, as mentioned in our answer to your Q3.
>
>
> Q2: **Real-world simulated tasks**
> Thanks for your discussion on the simulated tasks. To clarify, by “show a clear contribution to planning”, we mean that, at the current stage, we intend to use pure planning tasks to show the LLM planning performance, with PDDL and real-world motivated natural language planning.
>
> Compared to PDDL or trip planning, real-world simulated tasks, as one kind of planning task, are compound, i.e., agentic. Besides the planning-side capability, in simulated tasks, the LLM-based agents are also additionally required to capture and understand the complex environmental feedback per action (more discussion can be found in Slide 18 of a recent tutorial by the authors of Mind2Web: https://tinyurl.com/language-agent-tutorial-2024).
>
> In a recent work [1] in Nov 2024, they show how it is hard for LLMs to simulate outcomes without special design in the pipeline. Both the planning and environment understanding are required for the simulated tasks and are in active investigation of the community.
> If ICL performance is bounded by the model's ability to simulate the outputs from HTML (Mind2Web) or a specific multi-modal API server (ALFWorld), the contribution on the planning side will be blurred.
>
> Let’s further discuss the difference with examples. As shown in the example of trip planning, Appendix A.1, all the information needed for the planning task is clearly given in the descriptions. The understanding of the “dynamic environment” is bounded in the descriptions. “Fly from Helsinki to Barcelona” is a planned action to satisfy the constraints, but there will be no feedback from the actual simulated execution, e.g., updated luggage information when arrives.
> However, for simulated tasks, the interactions between agents and the dynamic environment are not fully disclosed to the LLMs in the task descriptions, e.g., for Mind2Web, as shown in [1], clicking “Electronics” will need a refreshed sub-menu with different displayed products, i.e., requiring feedback utilization as defined in [9].
>
> Thus while simulated tasks offer a valuable testing ground, they require LLM-based agents to go beyond planning and delve into interpreting complex environmental feedback. This aspect, however, falls outside the scope of our paper.
>
> In general, automated planning has long been studied as a problem with various research insights and real-world applications [2] and recently with LLMs [3]. Studying the planning capability is important and has implications over various downstream tasks besides simulated tasks, e.g., real-world embodied reasoning [4], coding [5][6], and math [7]. PDDL is commonly accepted as a good and controlled setting to evaluate LLM planning [3][7][8].

---

> ### Author Response · Authors · 2024-11-27
> **Continue our comments...**
>
> References
>
> [1] Is Your LLM Secretly a World Model of the Internet? Model-Based Planning for Web Agents
>
> [2] Automated Planning: Theory and Practice, 2004, https://www.google.com/books/edition/Automated_Planning/eCj3cKC_3ikC?hl=en&gbpv=0
>
> [3] Position: LLMs Can’t Plan, But Can Help Planning in LLM-Modulo Frameworks
>
> [4] Inner Monologue: Embodied Reasoning through Planning with Language Models
>
> [5] Planning In Natural Language Improves LLM Search For Code Generation
>
> [6] CodePlan: Repository-Level Coding using LLMs and Planning
>
> [7] Reasoning with Language Model is Planning with World Model
>
> [8] Leveraging Pre-trained Large Language Models to Construct and Utilize World Models for Model-based Task Planning
>
> [9] AutoPlan: Automatic Planning of Interactive Decision-Making Tasks With Large Language Models
>
> Q3: **Paper modification**
> Thanks so much for the suggestions. Following these bullets, we have uploaded a new version of the paper. By “final version”, we mean a refined version considering all rounds of discussion during the rebuttal period. The modified content is marked in blue.
> Specifically, we (1) add discussion on VAL and edit the typo in Sim_AS in Section 2.1; (2) refer to the baseline AS as Basline_AS, instead of italic AS across the paper; (3) extend our explanation on +VAL in Section 2.2.1 and link it to rejection sampling (RS). In Appendix A.8, we further compared RS with random sampling and GRASE; (4) add a reference and discussion on Auto-CoT in Section 2.2.2 and Section 4.
>
> Finally, we appreciate your feedback. We hope our responses convince you about our methods and pure planning tasks. We kindly request that you review our responses and consider increasing your rating accordingly. If you have further questions or concerns, we are happy to continue our discussion.

---

### Official Review · Reviewer_gvwE · 2024-11-04

**Soundness:** 4
**Presentation:** 3
**Contribution:** 3
**Rating:** 8
**Confidence:** 4

**Summary:**

This paper proposes GRASE-DC, which is a new 2-stage pipeline for selecting examples for in-context learning of planning tasks based on action sequence similarity. In the ‘Generative Re-sampling of Action Sequence Similar Exemplars’ stage, GRASE-DC ranks and selects ICL exemplars with high action sequence similarities. In the ‘Dynamic Clustering’ stage, GRASE-DC performs AS-based clustering to further curate improved dynamic sets of examples that strikes a good balance between diversity and relevance to further improve the LLM’s planning ability. This paper also reports comprehensive experiment results on four PDDL tasks to empirically show the performance gain brought by GRASE-DC to different LLMs.

**Strengths:**

- **Originality**: This paper has good originality in that it proposes to focus on action sequence similarity instead of the traditional criteria based on the semantic similarity between task descriptions when performing example selection for LLM in-context learning of planning tasks.
- **Quality**: This paper has high overall quality. Most of the steps in the proposed GRASE-DC pipeline are very clearly described and discussed in the methodology section, and Section 3 as well as the appendix also provide comprehensive empirical results and analysis to support the main claim of the paper.
- **Clarity**: This paper is generally well-written and the ideas are generally well-presented in the paper. The clarity of both the writings and the figures are good. Please see the Weaknesses section in this review for things to improve.
- **Significance**: The main intuition behind GRASE-DC is relatively simple and straightforward, but its performance gain shown in the empirical evaluations is solid.

**Weaknesses:**

1. In the formula on Line 141, shouldn’t there be a ‘|      |’ symbol around ‘LCAS(A_i, A_j)’? According to the previous description, LCAS(A_i, A_j) is a sequence, not a number.
2. The notation and definition of the core concept ‘Action Sequence Similarity’ should be defined more clearly and strictly in the paper. Currently some mentions of AS are a little vague and confusing.

**Questions:**

In Figure 3, why on some tasks the planning accuracies of GRASE+VAL are higher than those of AS for certain numbers of examples? Is there any intuition behind this observation?

---

> ### Author Response · Authors · 2024-11-19
> **Thank you for your review.**
>
> Thank you for your valuable and detailed feedback. Thanks for pointing out the typo, we will add the || symbol around LCAS as we did on the components in the denominator.
>
> Regarding the notion, in the paper, we use AS to denote the similarity scores and the italic version to denote the corresponding analytical baseline. We sincerely appreciate your suggestion for improving the clarity. We will use Sim_AS to refer to the scores and AS to refer to the baseline using plan similarity in the final version of the paper.
>
> Regarding the intuition behind the performance, extending the discussion in lines 409-417, we assume that one potential reason is that, for AS, we use the reference plans for the test questions, however, there can be multiple potential plans satisfying the same test question. The model preference can be different from what is provided. In that case, sampling exemplars from the model-generated rough plans can potentially help the models refine their plans in the preferred direction. We recognize the systematic study of this feature would be an interesting future work upon the establishment of the proposed GRASE.
>
> Thank you for your positive remarks. We would be happy to answer any concerns/questions you may have about our submission.

---

### Official Review · Reviewer_p57G · 2024-11-04

**Soundness:** 3
**Presentation:** 2
**Contribution:** 3
**Rating:** 5
**Confidence:** 3

**Summary:**

The authors propose GRASE-DC, an approach to enhance planning capability in LLMs by improving ICL exemplar selection. Unlike traditional methods relying on semantic similarity, which can lead to false positives, GRASE-DC utilizes AS similarity to identify exemplars with closely aligned plan structures. The GRASE-DC pipeline comprises two stages: re-sampling high-AS exemplars and dynamically clustering them to balance relevance and diversity, yielding more accurate planning guidance with fewer exemplars. An iterative variant, GRASE-DC*+VAL, further boosts performance.

**Strengths:**

1. The proposed approach is both intuitive and shows good empirical performance as it selects exemplars based on AS similarity, providing the similar types of exemplars as the test task.
2. The empirical evaluation is extensive, spanning four PDDL tasks and a natural language planning task, and it tests the method across different base models, showcasing the robustness of the approach.

**Weaknesses:**

1. The effectiveness of GRASE is highly dependent on the quality of initial plans generated by the LLM with randomly selected exemplars; poor initial plans can lead to compromised AS-based exemplar selection.
2. For setups with validator access, a baseline comparison with rejection sampling could improve the analysis. E.g., under a similar validator query budget, the validator can be used to reject the invalid plans and select the better plan generated by the approach with random exemplars.

**Questions:**

Refer to weaknesses.

---

> ### Author Response · Authors · 2024-11-19
> **Thank you for your review.**
>
> Thank you for your valuable and detailed feedback. We address your questions below.
>
> Q1: **Reliance on the initial plan quality.**
> Thanks for the comments. As mentioned by the reviewer, the performance on the first iteration of GRASE can be influenced by poor initial plans. Yet, interestingly our results confirm that the iterative application of GRASE can relieve this problem. We show the performance of different domains in Figure 3, we can observe that some domains have high initial model performance (e.g., around 40% accuracy on Blocksworld) and some do not (e.g., less than 10% accuracy on Tetris). As shown in the upper row of Figure 3, we can observe that there is a +10% accuracy improvement with the first iteration of GRASE, though the overall performance is 15% due to the low initial performance, despite the great improvement. However, as shown in the lower row, through one more iteration, we can observe that GRASE-DC (iteration 2) achieves around 45% accuracy on Tetris planning, which is much better than the initial performance. The issue of random initialization can be relieved with iterated generation with plans from a previous GRASE step that has higher performance.
>
> Q2: **Rejection Sampling.**
> Thanks so much for the thoughtful suggestion on the rejection sampling baseline. As suggested we ran new experiments and compared iterative rejection sampling (-RS) with exemplars selected from Random (Rand.) or by GRASE on Blocksworld. Similar to VAL, we can also iteratively apply rejection sampling, denoted as -RSx2. For GRASE, -RS equals +VAL in the original paper. The performance is as follows (best entries are in bold):
>
> | Method/# exemplars | 1 | 4 | 10 | 20 | 40 | 100 |
> | :-------------------: | :-: | :-: | :-: | :-: | :-: | :-: |
> | Rand. | 36 | 39 | 43 | 37 | 37 | 35 |
> | GRASE | 51 | 58 | 55 | 53 | 50 | 49 |
> | Rand.-RS | 52 | 53 | 54 | 52 | 49 | 49 |
> | GRASE-RS | 64 | 64 | 59 | 57 | 55 | 54 |
> | Rand.-RSx2 | 59 | 59 | 59 | 49 | 49 | 54 |
> | GRASE-RSx2 | **70** | **66** | **66** | **67** | **69** | **69** |
>
> From the table, we can observe that rejection sampling (assuming we have the gold validator) helps both random and GRASE. However, GRASE-RS shows a 5-12 point accuracy improvement over Rand.-RS. On the second iteration, the gap between Rand and GRASE becomes larger, which validates our performance gain. We will include the new baseline and discussion in the final paper.
>
>
>
> Thank you for your time and effort in reviewing our paper. We have incorporated your suggested experiments and believe they have strengthened the paper. We hope you increase your rating in light of these revisions. Please let us know if you have any further questions.

---

> > ### Comment · Reviewer_p57G · 2024-12-03
> > **Response to rebuttal**
> >
> > Thank you for your rebuttal. The argument for Q1 is valid, and I agree that the problem of low-quality initialization can be mitigated by iterative refinement of the in-context exemplars. Regarding Q2, the intent is not to discuss where RS can improve your approach or random. Instead, I seek to understand how RS performs in comparison to GRASE under an equivalent validator query budget. I assume `Rand.-RS` and `GRASE-RS` are not directly comparable as GRASE needs extra validator queries.

---

> ### Author Response · Authors · 2024-12-01
> **Thanks again for your review...**
>
> Thank you for your comments. We truly appreciate your detailed review, which has been invaluable in improving our work. In the last round of responses, we discussed the robustness of the improvement and added experiments on rejection sampling (Appendix A.8 in the current revised draft). We would greatly appreciate it if you could share any further insights or suggestions. Otherwise, if you feel that we have satisfactorily addressed your concerns, we kindly ask: if you can consider increasing your score before the end of the discussion to support our paper's acceptance. Thanks!

---

> ### Author Response · Authors · 2024-12-03
> **Thank you for the discussion**
>
> Thank you so much for your response! We are so glad that you acknowledge our argument for Q1 regarding mitigating low-quality initialization.
>
> Regarding Q2, in our current implementation, Rand.-RS and GRASE-RS are comparable since the rejection sampling is done by sending requests to PDDL VAL, which is commonly a package that can execute plans written in PDDL, e.g., one implementation: https://github.com/KCL-Planning/VAL. For both Rand./GRASE-RS, we generate plans from different exemplar sampling methods two times and mark a success if there is one plan achieves the goal. For each test question, both methods query the LLMs twice and send 2 requests to the validator. With GRASE, we strategically sample the exemplars with the generated plans, yet this additional computation on AS requires no query to the LLM/validator. For GRASE (as well as the iterated version, GRASE-DC*, all green lines in Figure 3 of our draft), we re-sample the exemplars for all the cases without knowing if the previous step plans are correct. Only with +VAL/RS, do we assume we get that information from PDDL VAL.
>
> We also consider using queries to let LLMs simulate PDDL VAL, with the domain descriptions, task, and intended plan as the context. The agreement between the LLM validator and Oracle rule-based PDDL validator is low (77% agreement on whether a plan is correct), which indicates that there is further effort needed in future work to use LLM-based queries to simulate rule-based PDDL. We are also excitedly looking forward to this new direction following our current work.
>
> Thanks again for your valid suggestions and for engaging in the discussion! Please let us know if you have further concerns and we can work on addressing them before the approaching deadline. If you feel we have satisfactorily addressed your concerns, we would sincerely appreciate it if you could consider increasing your rating.

---

### Author Response · Authors · 2024-12-04
**Thanks for the discussion period**

We express our sincere gratitude to the area chair and reviewers for hosting an insightful and constructive discussion.

We appreciate that the reviewers acknowledge that:

1. The paper is well-written with a clear presentation of the ideas (Reviewer gvwE, BrrL)

2. The proposed method from simple and effective intuition shows significant empirical performance on planning tasks (Reviewer p57G, gvwE, BjER, BrrL)

3. The empirical evaluation is extensive, showcasing the robustness of the approach across backbone LLMs and settings (Reviewer p57G, BjER, BrrL)

Through the discussion, we have carefully addressed and responded to the concerns and questions raised by the reviewers, as acknowledged in the discussion, which are summarized as follows:

1. In response to Reviewer p57G, as acknowledged by the reviewer, we first describe how results in the original draft (good performance on Tetris planning) can address the concerns about LLMs’ initial performance. We then conduct additional experiments to address the reviewer’s concern about the rejection sampling baseline and show robust gain from GRASE. In the second round of discussion, we explained how the rejection sampling with random and GRASE is comparable. We believe the discussion and new results can provide a good response to the reviewer’s concern.

2. In response to Reviewer gvwE. We sincerely appreciate the reviewer for liking our paper and we updated the formula and notion of action sequence similarity in the paper.

3. In response to Reviewer BjER, in the first round of discussion, we addressed the reviewer’s confusion with PDDL validators, efficiency comparison, and the iteration decision of the proposed methods. We also further discussed the details of the significance of our work compared to the literature. In the second round of the discussion, we further introduce how GRASE-DC is a point-wise (i.e., per test question) clustering method on exemplar selection with signals from planning, which differs from previous work on QA tasks with a single fixed cluster. We also further show the significance of our test suite (4 pure planning tasks and 2 natural planning tasks) with abundant previous work in the communities and detailed comparison with simulated tasks with examples.

4. In response to Reviewer BrrL, in the first round of discussion, as acknowledged by the reviewer (“The clarifications and new empirical results have addressed some of my earlier concern”), we successfully convinced the reviewer of the significance of our ICL-based approach, new baselines from plan-side similarity, and pseudo code to show the details of the implementation.
In the second round of discussion, we further answer the reviewer’s additional questions on the exact latency and how our method shows great performance with a multiple iteration variant (i.e., GRASE-DC*) in the original paper. At this point, we have not heard the reviewer’s further insights or details. We believe the current answers provide a good response to these questions.

Accordingly, in light of the reviewers' great suggestions, we improved our draft to incorporate the discussion, as mentioned in our response to Reviewer BjER, which is summarized as follows:

1. We add discussion on PDDL VAL and edit the typo in Sim_AS in Section 2.1;
2. We update the notion of action sequence similarity (AS) as we refer to the baseline AS as Basline_AS, instead of italic AS across the papers;
3. We extend our explanation on +VAL in Section 2.2.1 and link it to rejection sampling (RS).
4. In Appendix A.8, we further compared RS with random sampling and GRASE;
5. We add a reference and discussion on Auto-CoT in Section 2.2.2 and Section 4.

As acknowledged by the reviewers, our method shows a solid contribution to improving the planning performance of large language models from the novel perspective of plan similarity. The simple yet effective intuition we showcase can also inspire further ideas in the community. We believe that our responses and corresponding changes can address most of the reviewers’ concerns and questions. The new points in the second round discussion are straightforward to be integrated into the final draft.

We sincerely appreciate the area chair's effort in hosting the discussion period and the reviewers' collaborative effort and engagement. We hope that our contributions and responses can be taken into full consideration.

---

### Meta-Review · Area_Chair_q68i · 2024-12-24

**Metareview:**

The paper proposes a method to enhance LLM’s planning by improving the example selection in in-context learning. Instead of leveraging semantic similarity, the proposed method uses AS similarity. The writing is mostly easy to follow. The AS similarity is intuitive, and the performance improvement is grounded. As the paper suggests AS similarity is more suited for planning tasks, a running or illustrative example would help better concretize the idea. The method is sensitive to its initial plan (which is affected by randomly selected examples). The added analysis and ablation study in the rebuttal period makes the comparison more insightful and more grounded.

**Additional Comments On Reviewer Discussion:**

Most concern lies on the experiments. Some baselines with straightforward adaption should be added to make the comparison more fair and the analysis more grounded. The concern is addressed and most reviewers are convinced during the discussion period.

---

### Decision · Program_Chairs · 2025-01-22

Accept (Poster)